# DeepCristae, a CNN for the restoration of mitochondria cristae in live microscopy images

Salomé Papereux [1,2,5], Ludovic Leconte[1,2,5], Cesar Augusto Valades-Cruz [1,2,5], Tianyan Liu[3], Julien Dumont[4], Zhixing Chen [3], Jean Salamero [1,2], Charles Kervrann [1,2] & Anaïs Badoual [1,2] ✉

Mitochondria play an essential role in the life cycle of eukaryotic cells. However, we still don't know how their ultrastructure, like the cristae of the inner membrane, dynamically evolves to regulate these fundamental functions, in response to external conditions or during interaction with other cell components. Although high-resolution fluorescent microscopy coupled with recently developed innovative probes can reveal this structural organization, their long-term, fast and live 3D imaging remains challenging. To address this problem, we have developed a CNN, called DeepCristae, to restore mitochondria cristae in low spatial resolution microscopy images. Our network is trained from 2D STED images using a novel loss specifically designed for cristae restoration. To efficiently increase the size of the training set, we also developed a random image patch sampling centered on mitochondrial areas. To evaluate DeepCristae, quantitative assessments are carried out using metrics we derived by focusing on the mitochondria and cristae pixels rather than on the whole image as usual. Depending on the conditions of use indicated, DeepCristae works well on broad microscopy modalities (Stimulated Emission Depletion (STED), Live-SR, AiryScan and LLSM). It is ultimately applied in the context of mitochondrial network dynamics during interaction with endo/lysosome membranes.

The study of certain pathologies has shown the importance of mitochondria, which above all, ensure ATP production within cells and are central in many biological functions (e.g., metabolic pathways, ion homeostasis, apoptosis, autophagy, epigenetics…)[1,2]. Mitochondrial energetic adaptations to environmental constraints encompass a plethora of processes that maintain cell survival. An alteration of these processes generally leads to serious diseases such as cancer, neurodegenerative and cardiovascular disorders[3]. Although much attention has been paid to the role of mitochondria, the precise niche the organelle plays in cell life and death still remains unclear. The lack of in-depth knowledge about the ultrastructural evolution of mitochondria in live cells, under normal and stressful conditions, might be one of the blind spots. In particular, the cristae formed by the inner membrane of mitochondria that concentrate ATP production in a defined area, their dynamic behavior, sublocation or density have been poorly related to the various functionalities or dynamic processes

(e.g., fusion, fission) that mitochondria undergo. The challenge we address lies in imaging mitochondria cristae, which measure between 30 and 50 nm wide[4], at a high spatial and temporal resolution so that their structural dynamics and interactions can be accurately studied over time for several dozens of milliseconds to a few seconds. However, this is starting to be possible with the recent development of high-resolution imaging approaches[5].

Stimulated emission depletion (STED) microscopy, which allows for sub-diffraction resolution (*xy*: 30–50 nm), is one of the very few techniques[6,7] able to decipher dynamics of mitochondria cristae in live cells[4]. However, their observation in 3D and in fast time is limited by the acquisition frame rate capacity (1 plane ≈ 1 to 10 s). In addition, depletion STED, which is the principle that achieves nanoscopic resolution, induces local heat by high illumination intensity[8] to which mitochondria are known to be particularly sensitive[9,10]. This can affect their overall physiology and potentially lead to

[1]SERPICO Project Team, Centre Inria de l'Université de Rennes, Rennes, France. [2]SERPICO Project Team, UMR144 CNRS Institut Curie, PSL Research University, Paris, France. [3]College of Future Technology, Institute of Molecular Medicine, National Biomedical Imaging Center, Beijing Key Laboratory of Cardiometabolic Molecular Medicine, Peking-Tsinghua Center for Life Science, Academy for Advanced Interdisciplinary Studies, Peking University, Beijing, China. [4]CIRB Microscopy Facility, Collège de France, UMR 7241 CNRS, Inserm U1050, Paris, France. [5]These authors contributed equally: Salomé Papereux, Ludovic Leconte, Cesar Augusto Valades-Cruz. ✉e-mail: anais.badoual@inria.fr

apoptosis and mitophagy. A number of new fluorescent probes that are more photostable with less saturation intensity and that allow cristae decoration, have been developed in the very last years[7,9,11,12]. Yet, the application of a dark recovery step ($\approx 30$ s) after STED imaging is still necessary, again at the expense of temporal resolution. This could be improved by applying a partial STED depletion protocol, leading to an intermediate quality resolution ($xy \approx 100$ nm)[13], but insufficient to spatially resolve mitochondria cristae and not solving the frame rate limitation (4–5 s in average).

In this context, one solution to study the dynamics of mitochondria cristae is to collect as much temporal information with minimal photo-toxicity using an appropriate microscope, and then restore the spatial dimension using computational methods. Indeed, the development of image restoration algorithms has become increasingly popular in recent years with the need for nanoscale analysis[14–23]. At the heart of fluorescence microscopy have been actively developed denoising algorithms[24–29], dedicated to images corrupted by a mixed Poisson-Gaussian noise, as well as deconvolution algorithms[30–32], designed to remove the blur induced by the limited aperture of the microscope objective. Some methods combine the two approaches[33]. However, these conventional restoration methods usually rely on general assumptions, such as the nature and level of noise and spatial regularity, which hampers their effectiveness on the diversity of structures and level of degradation in microscopy images. Over the years, the literature on image restoration has evolved considerably due to deep learning and the rapid growth of convolutional neural networks (CNNs). These methods have the advantage of making assumptions based on image content, resulting in state-of-the-art performance in denoising[34,35] and deblurring[14,19,22,23,36–40] fluorescence microscopy images. However, these methods have two major drawbacks. First, these CNNs often require a training step based on a large ground truth dataset that is generally not available in microscopy. Second, they focus on restoring the entire image, while sometimes little information is worth restoring within it, especially in the dark background. This is the case with mitochondria cristae, which have a sparse number of pixels in the image compared to the background. Therefore, CNNs that have been previously applied to mitochondria microscopy images[21,22,39,41] provide good global restoration of the background and mitochondria but fail to accurately restore fine details as cristae, especially in very low spatial resolution images. To circumvent this, new conventional methods have been proposed to enhance resolution and suppress artifacts in high-resolution techniques, including Hessian-SIM[17]. However, the denoising results are limited when dealing with low signal-to-noise ratio images and Hessian deconvolution assumes that the unknown image is smooth and sparse. A hybrid solution has been proposed in TDV-SIM[42], which combines the strengths of conventional physical model-based algorithms with deep learning-based algorithms. Another hybrid solution, rdLSIM[21], incorporates the deterministic physical model of specific micro-scopy into network training and inference. Nevertheless, the effectiveness of these methods, along with conventional restoration algorithms, relies on the careful selection of optimal parameters or on prior knowledge of illumination patterns, respectively.

Instead of developing an additional generic image restoration method that may not satisfactorily enhance certain sparse but informative pixels in the image, we present DeepCristae, a CNN specifically developed to restore mitochondria cristae in low spatial resolution microscopy images. DeepCristae was applied to several microscopy modalities and different biological scenarios capturing live mitochondria at high speed with low illumination and thus low phototoxicity. DeepCristae allows long-term/fast dynamic observation of cristae behavior and organization. The main challenge was to handle the low number of cristae pixels compared to the background in the acquired images. Therefore, the main contributions of this work are (1) the design of a new training loss dedicated to the restoration of specific pixels of interest, (2) the development of a random image patch sampling focusing on areas of mitochondria to increase the size of the training set, and (3) the building of metrics for objective assessment of cristae restoration.

## Results

### Overview of DeepCristae

DeepCristae aims to restore mitochondria cristae in intermediate to low spatial resolution microscopy images. Its pipeline is illustrated in Fig. 1. DeepCristae mainly consists of a U-Net trained on a dedicated dataset built from real high-resolution 2D STED images (Methods) and using a novel training loss we specifically designed for cristae restoration (Methods, Eq. (1)). Although the term is not fully appropriate, for simplification we refer to this dataset as "synthetic" $D_{synt}$. A pipeline for random image patch sampling focusing on regions of mitochondria in the acquired data was also developed (Methods) to efficiently increase the size of the training set of $D_{synt}$ and avoid empty patches. DeepCristae image restoration network was implemented in Python (TensorFlow version 2.11) and is freely available as an open-source software (see code availability). DeepCristae is also integrated into BioImageIT[43], an open-source platform with existing software for microscopy.

### DeepCristae quantitatively outperforms state-of-the-art algorithms on the synthetic dataset $D_{synt}$

Our method was quantitatively compared to existing both conventional (Richardson-Lucy[30,31], Wiener[32], SPITFIR(e)[33]) and deep learning (ESRGAN[40], CARE[19], RCAN[38], and SRResNet[36]) algorithms for image restoration. Details about their implementation are in Supplementary Note 2.2.2. All deep learning methods were trained from the same patches extracted from the training set of $D_{synt}$. To evaluate the performance of the different methods, we used current metrics, namely normalized root mean square error (NRMSE), peak signal-to-noise ratio (PSNR) and structural similarity index (SSIM) (see Supplementary Note 2.1). However, these measures are relevant to the image as a whole, but insufficient in the context of mitochondrial cristae restoration. Indeed, the images contain only a few pixels of cristae and thus have too little impact in those metrics unlike the many background pixels. To overcome this issue, we encouraged the evaluation metrics to focus exclusively on mitochondria pixels (Supplementary Fig. 1, second column). We call these mitochondrial metrics $NRMSE_{mito}$, $PSNR_{mito}$, and $SSIM_{mito}$. To go one step beyond and accurately assess cristae restoration, we also introduced the cristae metrics $NRMSE_{cristae}$, $PSNR_{cristae}$, and $SSIM_{cristae}$. These metrics are computed over mitochondria cristae pixels only, obtained from manual annotations (Supplementary Fig. 1, third column). More details about these customized metrics are given in Supplementary Note 2.1. Each competing algorithm was evaluated over the test set of $D_{synt}$ for the nine aforementioned metrics (Fig. 2a). For all measurements focusing on cristae, DeepCristae ranks first, and is either first or second otherwise. Conventional methods behave worse than deep learning approaches, CARE appearing as DeepCristae's most competitive method. In terms of visual assessment, we make the same observation (Fig. 2b). RCAN amplifies the background noise, resulting in less accurate restoration of cristae and unrealistic reconstructed structures in the background or in mitochondria. DeepCristae and CARE remove noise background while restoring most of the cristae details. However, CARE restores mitochondria cristae with less sharpness compared to DeepCristae, especially for mito-chondria with low contrast (Fig. 2b, CARE white arrows). This improvement by DeepCristae is highlighted by the values of the metrics $NRMSE_{cristae}$, $PSNR_{cristae}$, and $SSIM_{cristae}$, and by the Fourier Image REso-lutions (FIREs) computed using Fourier Ring Correlation Plugin[44] (Fig. 2c). We also demonstrated that DeepCristae outperforms CARE by quantitatively studying their performance in terms of cristae resolution (Fig. 2d–f). We measured cristae widths for 155 cristae (mean of 92.44 ± 23.59 nm on HR STED) from the test set of $D_{synt}$ by fitting line profiles (Fig. 2d) to a Gaussian model and measuring the Full Width at Half Maximum (FWHM) (Supplementary Note 1.3). DeepCristae slightly improves the number of cristae restored compared to CARE and, on average, restores individual cristae at 137.62 ± 59.64 nm of resolution, as compared to 143.15 ± 71.15 nm for CARE (Fig. 2e). This improvement is statistically relevant as confirmed by the results shown in Fig. 2f.

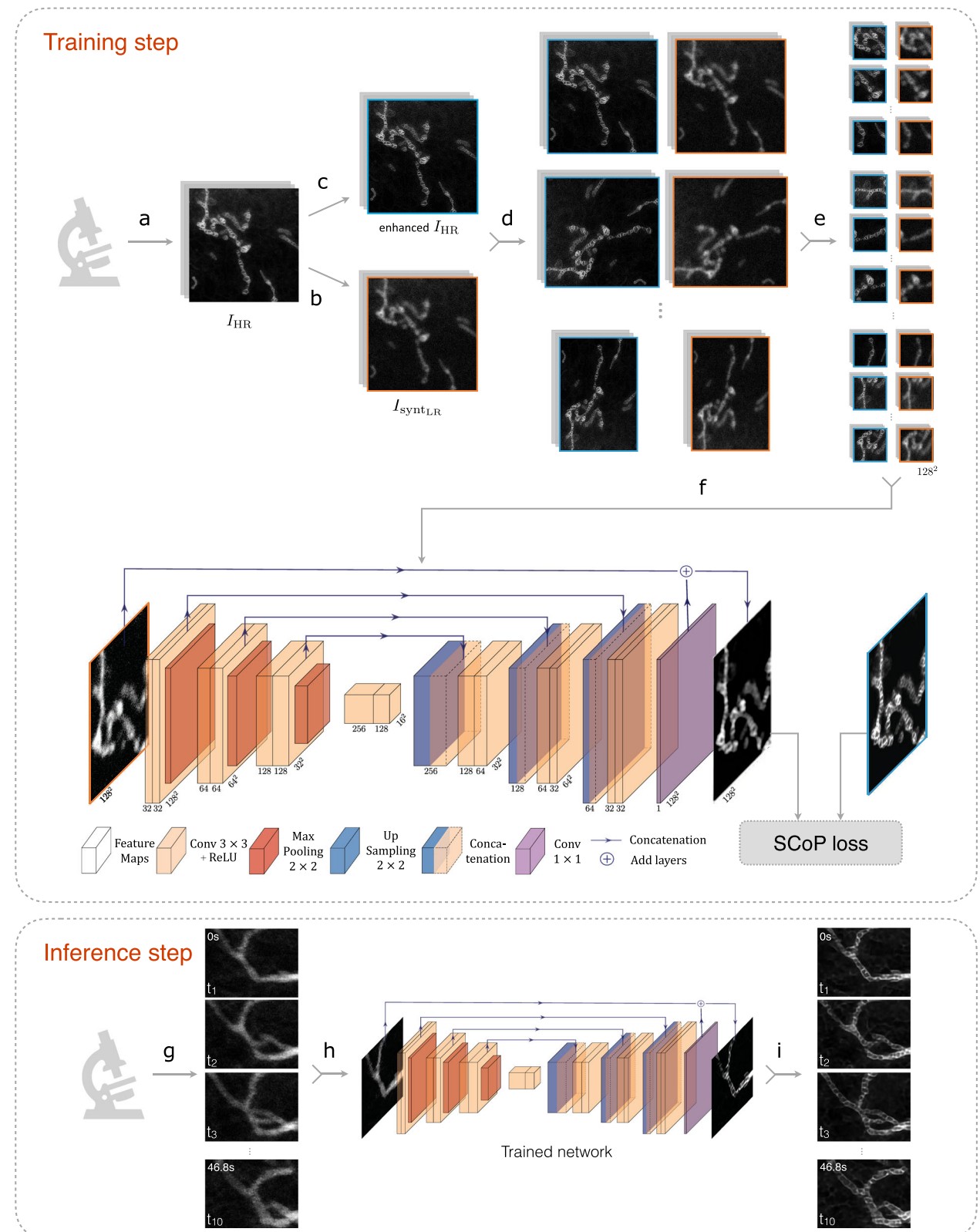

In Fig. 3, we take a closer look at three of the restorations previously obtained by DeepCristae on $D_{synt}$. For each restoration, a comparison of normalized intensity profiles is performed between the input image, the DeepCristae restored image, and the high-resolution (HR) STED image. It shows that DeepCristae restores spatial information by revealing mitochondria cristae while improving signal to noise ratio.

## Robustness of DeepCristae with respect to noise, blur and mitochondria scale in the low-resolution images

We have shown that DeepCristae performs well on 2D STED images and outperforms state-of-the-art algorithms. However, it is important to verify the reliability of DeepCristae more widely. DeepCristae has been trained on a dedicated dataset acquired with specific microscope settings and

**Fig. 1 | Overview of DeepCristae.** *Training step*: (**a**) Acquisition of 33 high-resolution (HR) 2D STED images of live RPE1 cells stained with PKMITO-Orange for mitochondria. From these HR images ($I_{HR}$), counterpart low-resolution (LR) images ($I_{synt_{LR}}$) were created (**b**) to form the dataset $D_{synt}$: resolution degradation of the $I_{HR}$ images by applying Gaussian filtering (with standard deviation $\sigma_{blur} = 3.25$ pixels) and by corrupting images with Poisson-Gaussian noise of standard deviation $\sigma_{noise} = 4.0$. **c** Enhancement of the mitochondria cristae on the $I_{HR}$ images using a Richardson-Lucy algorithm. The obtained dataset $D_{synt}$ is divided into a training set of 24 images and a test set. To increase the size of the training set, the pair of images $I_{synt_{LR}}/I_{HR}$ are then augmented (**d**) and sampled in patches of size $128 \times 128$ pixels

(**e**). We finally obtained 1824 pairs of HR images (blue) and LR input images (orange) to train our network (**f**), 80% of which is used for training and 20% for validation. The training is performed by minimizing our SCoP loss, especially dedicated to restoring mitochondria pixels. *Inference step*: (**g**) Long-term and fast acquisition with low illumination of live mitochondria. Note that if the training was performed on degraded STED images, the inference can be made on other microscopy modalities (e.g., Live-SR and Lattice Light Sheet Microscopy (LLSM)). **h**, **i** Frame-by-frame restoration of the acquired sequence by our previously trained DeepCristae network, allowing observation of the mitochondrial cristae dynamics at high resolution.

mitochondria properties (e.g., fluorescence markers, width in pixels of the mitochondria in the images). Any change in these settings during the inference step is expected to alter the quality of the restoration results. We performed experiments to evaluate the influence of changes in three parameters on the results: the level of noise, the amount of blur and the average width in pixels of mitochondria in the images to be restored. First, our model was trained on images obtained with specific parameters that mimic microscope settings: real images are assumed to be corrupted by mixed Poisson-Gaussian noise (with standard deviation $\sigma_{noise} = 4$) and the point spread function of the microscope is approximated by an isotropic Gaussian function of standard deviation $\sigma_{blur} = 3.25$ pixels. We investigated the robustness of DeepCristae to noise and to blur in the input images (Fig. 4). To that end, we corrupted the test images of $D_{synt}$ by several levels of mixed Poisson-Gaussian noise (from $\sigma_{noise} = 0$ to $\sigma_{noise} = 8$) and by different sizes of a Gaussian filter (from $\sigma_{blur} = 0$ to $\sigma_{blur} = 7$ pixels), independently. Note that these values of $\sigma_{noise}$ were chosen in line with the test images where the maximum intensity varies between 80 and 259. DeepCristae was applied to the resulting images and the metrics were computed (Fig. 4a, c). Visual results show that the quality of the restoration decreases as $\sigma_{noise}$ and $\sigma_{blur}$ increase (Fig. 4b, d). The higher the $\sigma_{noise}$ or $\sigma_{blur}$ values are, the blurriest the mitochondria's boundaries and their cristae. This is also confirmed by the evolution of the metrics as a function of $\sigma_{noise}$ (Fig. 4a) and by the evolution of $SSIM_{mito}$ and $SSIM_{cristae}$ as a function of $\sigma_{blur}$ (Fig. 4c). Surprisingly, the evolution of the $PSNR$ and $NRMSE$ as a function of $\sigma_{blur}$ have a bell-shape with a maximum and a minimum, respectively, for values of $\sigma_{blur}$ close to 3.25 pixels. We thus recommend using DeepCristae on microscopy images with blur and noise levels at worst equal to our training conditions ($\sigma_{noise} = 4$ and $\sigma_{blur} = 3.25$ pixels). Beyond this, the quality of the restoration can drastically decrease. Next, our model was trained from the training images of $D_{synt}$ depicting mitochondria of width $15.64 \pm 4.04$ pixels on average. We studied the quality of the predictions as a function of the mitochondria width in pixels in the input images. To that end, the test images of $D_{synt}$ were rescaled 11 times in order to contain mitochondria of specific widths (in pixels) on average. It thus results on 11 test sets on which our trained DeepCristae model was applied (Supplementary Fig. 2a, b) and the metrics were computed. The evolution of the metrics as a function of average mitochondrial width shows that the closer you get to the training parameters (i.e., an average width of 15.64 pixels), the better the quality of restoration. In fact, if the mitochondria are too small, few cristae are restored, and the mitochondria are thin. On the contrary, if the size is too large, DeepCristae tends to create artifacts looking like cristae patterns (Supplementary Fig. 2b, scaling of 31.28 pixels).

Finally, it is worth noting that DeepCristae has been developed and trained to restore mitochondria cristae in microscopy images. Consequently, any use of DeepCristae for other specimens or for any other application may lead to invalid results (Supplementary Fig. 2c, d).

### Reliability of image restoration by DeepCristae

It is important to guarantee that under well-controlled conditions of use (mitochondria width, image blur and noise), DeepCristae is stable and hallucination-free. By stable we mean that different training leads to consistent predictions. To demonstrate that this requirement has been met, we performed two experiments. First, we trained 10 DeepCristae neural networks with different training data (Fig. 5), each generated with our patch

generation method applied to the 24 training images of $D_{synt}$. For each training, the resulting model was applied to the test set of $D_{synt}$ and the metrics were computed. The average metrics obtained over the 10 trainings are close to the ones obtained with our model and the standard deviations are very low, showing consistency between predictions (Fig. 5a). By visually analyzing the predictions, the color map of the standard deviation (Fig. 5b) as well as looking at normalized intensity line profiles along mitochondria (Fig. 5c–f), we observe that the 10 trainings agree overall on the presence or absence of cristae but diverge in their intensity and their precise boundaries. In this experiment, all networks were initialized with the same weights, confirming that our image patch-sampling method is robust and leads to homogeneous learning. A second similar experiment was carried out. Ten trainings were performed from one dataset but with 10 different initializations of weights (Supplementary Fig. 3). This experiment indicates that the same dataset leads to homogeneous learning, meaning that the randomness of initialization does not play a key role in the learning process.

Now, that stability has been demonstrated, it is important to investigate if our method is hallucination-free. Correctly defining what hallucinations are and providing an appropriate quantifier is not trivial. In our case, it is reasonable to consider as hallucinations cristae perceived by DeepCristae that are non-existent (or imperceptible) in the data, creating nonsensical results such as cristae outside mitochondria or with a too high density inside mitochondria. To investigate hallucinations, we acquired 6 pairs of real 2D STED images. Each pair contains one low-resolution (LR) and one HR STED image, acquired as quickly as possible (~30 s), to minimize the displacements and deformations of mitochondria between the two acquisitions (Supplementary Note 1.2.1). The LR STED images were resized to have an average mitochondrial width of 15.64 pixels (391 nm), in line with the conclusions drawn above, and were then given as input to DeepCristae. The obtained predictions were qualitatively compared to the HR STED images to control their consistency (Supplementary Fig. 4). Four ROIs, from three of the six pairs of real STED images and corresponding predictions, were selected in regions where small mitochondrial displacements were observed to better appreciate the restoration (Supplementary Fig. 4a–c). For each ROI, a comparison of normalized intensity profiles between the input LR STED image, the DeepCristae restored image, and the HR STED image is performed (Supplementary Fig. 4d–g). Despite an offset due to mitochondrial motion, a correspondence can be established between the peaks of intensity, corresponding to cristae, of the line profiles of the restored images and the ones of the HR STED images. Moreover, the distance between cristae was measured along ten-line profiles taken through the six pairs of real data (Supplementary Fig. 4h). The metrics obtained on the restored images are comparable to the ones obtained on the HR images. Hence, a consistency in terms of location and density between the cristae restored by DeepCristae and the ones present in the HR STED images is observed overall. This result, combined with the stability of DeepCristae and the fact that no cristae were seen in the background, except for bad conditions of use (Supplementary Fig. 2b, scaling of 31.28 pixels), strongly suggests that we can consider DeepCristae hallucination-free, in the sense that if hallucinations exist, they are rare and minimal.

### DeepCristae enables to restore 2D+time STED images

To validate the restoration capabilities of DeepCristae in live biological samples, we first compare HR STED raw data with their DeepCristae

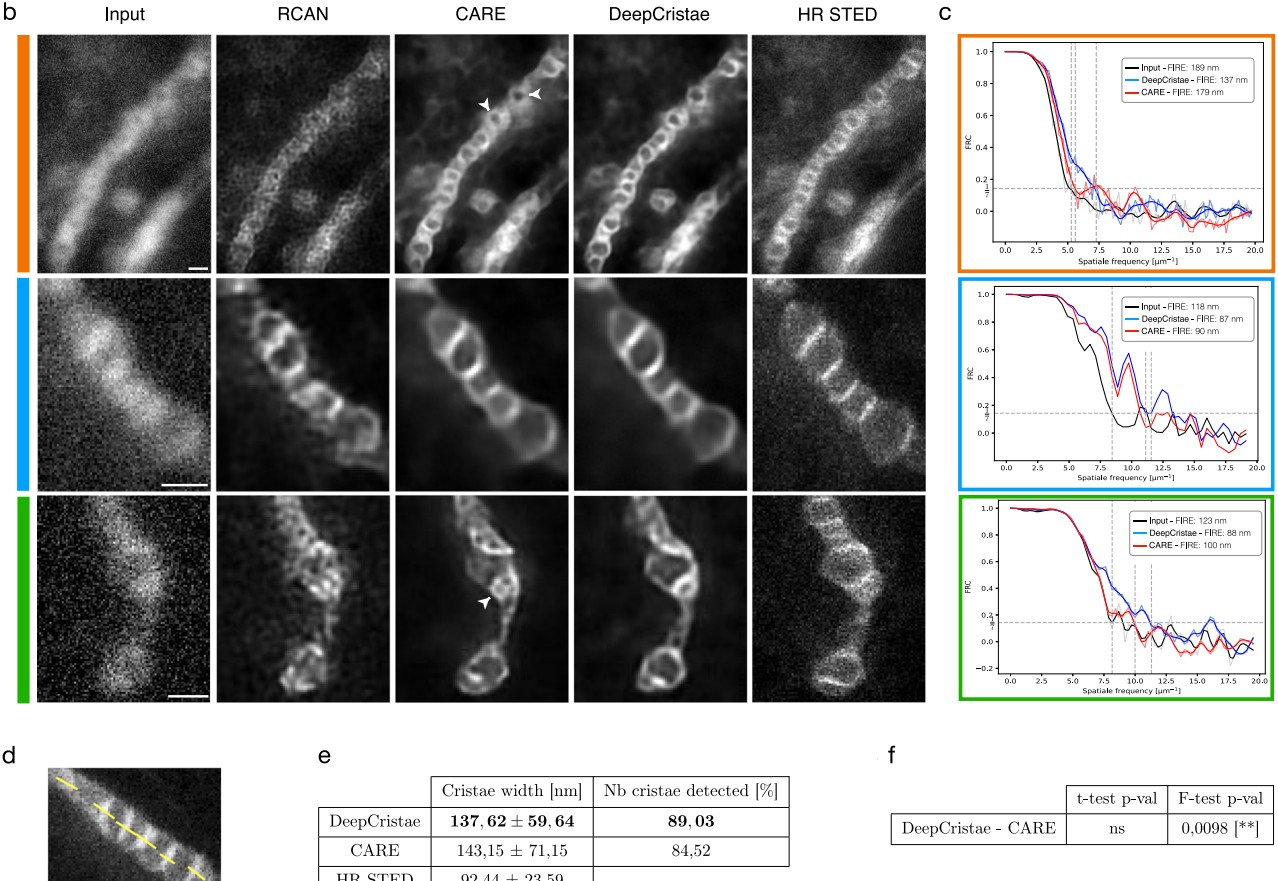

**Fig. 2 | DeepCristae outperforms state-of-the-art methods for restoring mitochondria cristae in low-resolution 2D STED images. a** Quantitative comparison of DeepCristae with conventional (Richardson-Lucy (RL)[30,31], Wiener[32], SPITFIR(e)[33]) and deep learning (ESRGAN[40], CARE[19], RCAN[38], and SRResNet[36]) image restoration methods. Metrics were computed on the test set of $D_{synt}$. Data are expressed as mean ± standard deviation. Note that all deep learning methods were trained using the same patches extracted from the training set of $D_{synt}$. Parameters used for conventional methods are indicated in Supplementary Note 2.2.2. **b** The image grid displays restoration results of 3 test images from $D_{synt}$ obtained with DeepCristae and two competitive deep learning methods: RCAN[38] and CARE[19]. Pixel size: 25 nm. Scale bar: 0.5 μm. White arrowheads indicate mitochondria with low contrast restoration by CARE; to be compared with DeepCristae column. **c** Fourier Image REsolution (FIRE) was estimated using Fourier Ring Correlation[44] for 3 test images before restoration, after CARE restoration and after DeepCristae restoration. **d–f** Measurement of cristae widths for 155 cristae from the test set after CARE and DeepCristae restoration. Line profiles (as depicted in (**d**)) were fitted to a Gaussian model and FWHM was measured (Supplementary Note 1.3). **d** Scale bar: 0.5 μm. **e** Table with the number of cristae restored by CARE and DeepCristae, in comparison to the 155 observed in HR STED images, and their average width. Data are expressed as mean ± standard deviation. **f** Table with statistical significance from Student's $t$-tests and Fisher's tests; ns non-significant.

restoration (Fig. 6a). This initial test confirms the absence of hallucinations, allowing us to proceed with a series of 2D+time data. While 2D STED nanoscopy enables to resolve mitochondria cristae and was here helpful to develop DeepCristae, live STED acquisition encompasses a number of hurdles. It includes relatively long-time frames between images, even when a photostable probe was used (Fig. 6b), limiting the temporal overview of the mitochondrial dynamics in the same plane. Moreover, STED imaging may rapidly induce photo-bleaching,

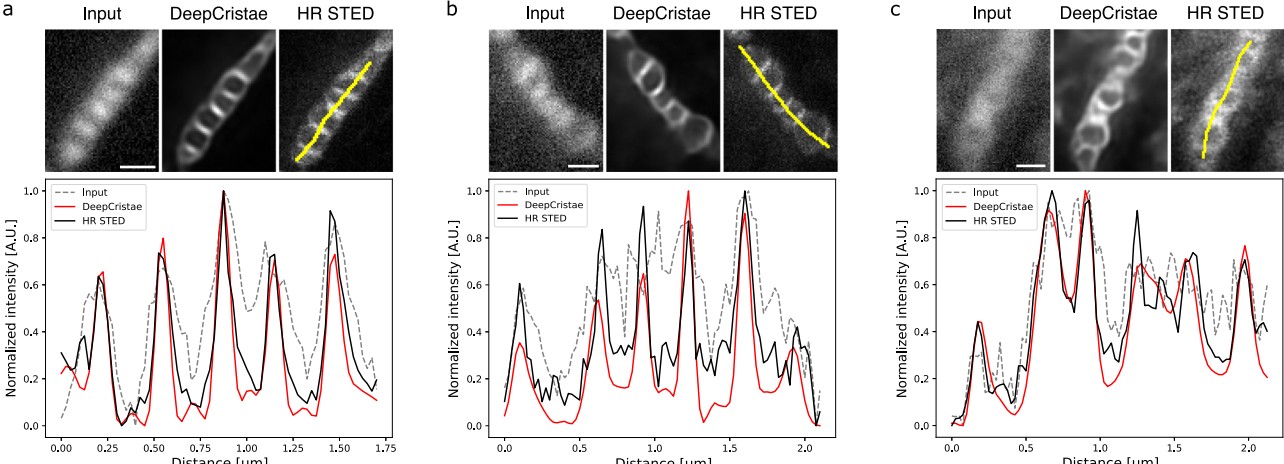

**Fig. 3 | DeepCristae reveals mitochondria cristae from low resolution (LR) 2D STED images. a–c** Restoration of 3 test images of $D_{synt}$ depicting RPE1 cells that were labeled with PKMITO-Orange for mitochondria. Pixel size: 25 nm. Scale bar: 0.5 μm. Top, from left to right: thumbnails of the LR image (Input), the image restored by DeepCristae and the HR STED image, respectively. Bottom: comparison of normalized intensity line profiles along a mitochondrion in the three thumbnails. The yellow line indicated in the HR STED thumbnail serves to identify the fluorescence profile.

which makes ultrastructural details progressively dimmed. More problematic, repeated STED imaging rapidly induces morphological deterioration of live mitochondria, illustrated by their swelling in the latest time points (Fig. 6b). This swelling effect was quantified here for HR STED by measuring the lateral widths of 7 distinct mitochondria over the 10 time points (Fig. 6d) in the image series (Fig. 6b). Swelling appears between the 4th and the 7th frames.

In order to improve the frame rate of 2D+time STED imaging while limiting photodamage on mitochondria, one may adjust the STED imaging protocols (Supplementary Note 1). Accordingly, we built another live STED dataset, first to indicate how long we can image under lower depletion conditions before the mitochondria are damaged, and second to control the efficiency of DeepCristae restoration over time. In what follows, these low-resolution (LR) STED images were referred to as Fast STED images. This goes at the dependence on the *xy* resolution (Fig. 6c, left bottom triangles in image series) for both the lateral width of mitochondria and the cristae width (Fig. 6d, Fast STED and Fig. 6e, Raw, respectively). Applying DeepCristae restoration on these latest series clearly revealed cristae morphology (Fig. 6c, right top triangle in image series). As expected, LR (or Fast) 2D+time STED images show little changes in mitochondria lateral widths in time, in contrast to HR STED (early and late time points in Fig. 6d), but a degraded resolution in the cristae widths (from a mean ($\mu$) of ~90 nm in HR STED to ~120 nm in Fast STED with standard deviation $\sigma = \pm 47$ nm). Applying DeepCristae allows recovery of a resolution lower than 100 nm and drastically reduces the variability of the measurement. The mean crista-to-crista distance, measured as peak-to-peak intervals (Fig. 6f), widely depends on the cristae density along the mitochondria network. Here, in RPE1 cell, it varies from 50 nm to 173 nm in early time points in HR STED ($\mu = 104.9$ nm, $\sigma = \pm 38$ nm) while the heterogeneity increases in late time points (from 130 nm up to 1.6 μm), consistently with the observable swelling of the mitochondria. In Fast STED, the cristae intervals measurements were non-significant. However, after DeepCristae restoration the mean crista-to-crista distance was estimated at ~142 nm ($\sigma = \pm 46$ nm) (Fig. 6f, g). Differences in these crista-to-crista measurements with similar studies[11] on HeLa or Cos7 cells for instance, will be further discussed. DeepCristae restored the individual cristae at 81 nm of resolution ($\sigma = \pm 9$ nm) at FWHM, as compared to the approximately 50 nm obtained in other studies[4]. DeepCristae provides a useful way to improve live STED nanoscopy by improving the resolution and decreasing the frame rates (3 to 6 s versus 13 s), yet with no observable photodamage as illustrated here by measuring the swelling of mitochondria.

## DeepCristae restores 3D+time images of mitochondria cristae by using intermediate high-resolution and diffraction limited microscopy

STED nanoscopy is not the only high-resolution microscopy adapted to resolve internal mitochondria ultrastructures in live cells. Indeed, a number of works using adaptation of SIM approaches have been published over the last few years[16,45], some combined with conventional deep learning methods[20,22,41]. Yet, the best compromise between fast and 3D imaging still remains an issue. We next investigated the performance of DeepCristae prediction on intermediate HR microscopies chosen for their 3D optical slicing performance. Spinning disk confocal equipped with a Live-SR module (or SDSRM for Spinning Disk Super Resolution Microscopy) is one of those well-disseminated systems equivalent to SIM. It improves the *xy* resolution by a factor of ~2 (~120–130 nm at 488 nm, ~140 nm at 561 nm)[46] while giving access to the depth (*z*-axis) of the sample and the live imaging of mitochondria (time t) without severe photo-bleaching and phototoxicity. The use of the Live-SR is therefore motivated here by both the study of these four dimensions and the ability of our model to correctly perform cristae reconstruction via multiple microscope imaging modalities. DeepCristae efficiently revealed cristae organization in single 2D Live-SR images acquired within 30 ms (Fig. 7a, upper images) and thus in 3D (Fig. 7a, lower images, MIP on 14 planes, with a stack time ~800 ms), giving access to the overall mitochondria network in the live cell at a fast rate. In this respect, it outperforms HR STED imaging and even Fast (LR) 2D STED imaging after DeepCristae restoration (compared to Fig. 6c). Cristae width comparative estimation (Fig. 7b) shows the improvement in resolution obtained after DeepCristae restoration on single plane Live-SR images (Raw: 149 ± 64 nm when measurable; DeepCristae: 87 ± 11 nm). These results are close to the expected widths of circumvoluted cristae tubules (50 to 100 nm) obtained by other methods derived from SIM imaging[22].

We then tested DeepCristae restoration on Lattice Light Sheet Microscopy (LLSM)[47] imaging which is not an HR microscopy by itself (at least in the dithered mode) but gives the best compromise in terms of fast 3D acquisition with minimal photon dose illumination and consequently low photo-damage of the mitochondria over time. Surprisingly, although with intrinsic limited and non-isotropic resolutions (PSF $xy = 300$ nm and $z = 600–700$ nm in our system) and a particular geometrical acquisition mode, cristae were however detectable in some mitochondria after realignment (deskew) and a Richardson-Lucy deconvolution. The resulting images were here considered as "Raw" data (Fig. 7c, 2D upper panel, left image). Applying DeepCristae to them (Fig. 7c, 2D

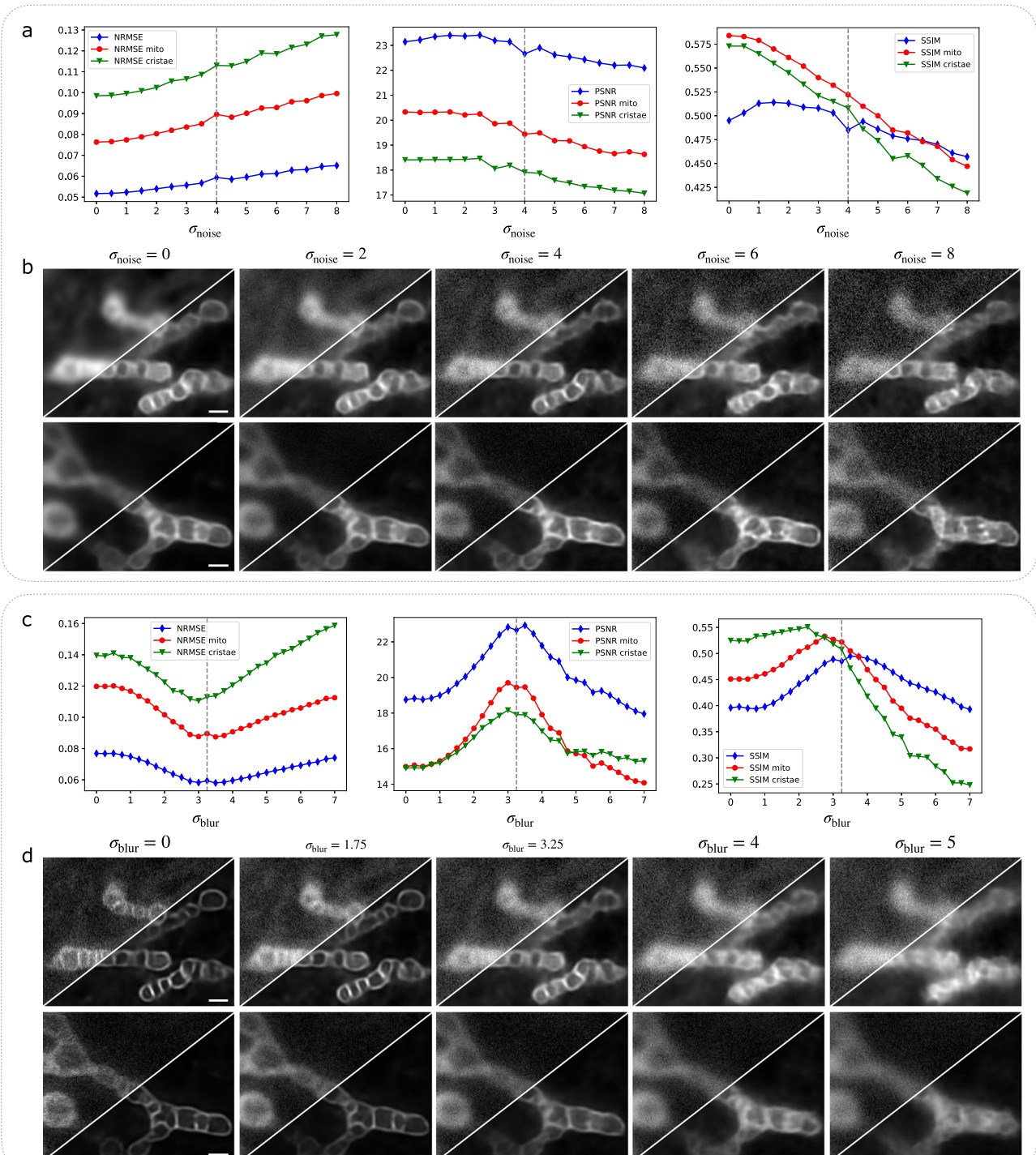

**Fig. 4 | Robustness of DeepCristae to noise and blur in the LR image.** *First experiment* (**a**, **b**): assessment of the robustness of DeepCristae to the level of noise in the image. To that end, RPE1 cells were labeled with PKMITO-Orange prior to 2D HR STED imaging. The obtained HR images, whose maximum intensity varies between 80 and 259, were then degraded with a Gaussian filter ($\sigma_{blur}$ = 3.25 pixels) to approximate the blurring effect due to the point spread function (PSF) of the microscope. Then, 17 test sets composed of 26 images were obtained by applying different levels of additive mixture of Poisson-Gaussian noise (from $\sigma_{noise}$ = 0 to $\sigma_{noise}$ = 8 with an increment of 0.5 unit, $\sigma_{noise}$ being the standard deviation of the Gaussian noise). Our trained model DeepCristae was then applied on each of these test sets. Note that our model was trained from the training set of $D_{synt}$ constructed with $\sigma_{blur}$ = 3.25 pixels and $\sigma_{noise}$ = 4. **a** Evolution of the metrics (*NRMSE*, *PSNR*, and *SSIM*) as a function of $\sigma_{noise}$. Full image metrics (blue line), mitochondrial metrics (red line), and cristae metrics (green line). The dashed lines on the plots indicate the training parameters of our model. **b** Illustrations of two test images for

five different values of $\sigma_{noise}$ (left to right: $\sigma_{noise}$ = 0, 2, 4, 6, 8) (top-left) and the corresponding predictions obtained with DeepCristae (bottom-right). Pixel size: 25 nm. Scale bar: 0.5 µm. *Second experiment* (**c**, **d**): assessment of the robustness of DeepCristae to the level of blur in the image. To that end, from the HR images described above, 29 test sets composed of 26 images were obtained by applying different sizes of a Gaussian filter (from $\sigma_{blur}$ = 0 to $\sigma_{blur}$ = 7 pixels with an increment of 0.25 pixel) to approximate different PSF sizes. Then, mixed Poisson-Gaussian noise with $\sigma_{noise}$ = 4 was added. Our trained model DeepCristae was then applied on each of these test sets. **c** Evolution of the metrics (*NRMSE*, *PSNR* and *SSIM*) as a function of $\sigma_{blur}$. Full image metrics (blue line), mitochondrial metrics (red line), and cristae metrics (green line). The dashed lines on the plots indicate the training parameters of our model. **d** Illustrations of two test images for five different values of $\sigma_{blur}$ (left to right: $\sigma_{blur}$ = 0, 1.75, 3.25, 4, 5 pixels) (top-left) and the corresponding predictions obtained with DeepCristae (bottom-right). Pixel size: 25 nm. Scale bar: 0.5 µm.

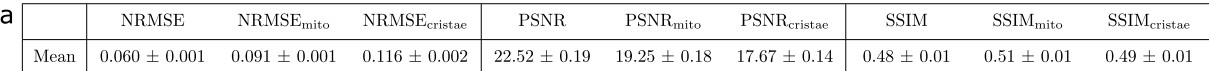

| | NRMSE | NRMSE_mito | NRMSE_cristae | PSNR | PSNR_mito | PSNR_cristae | SSIM | SSIM_mito | SSIM_cristae |
|---|---|---|---|---|---|---|---|---|---|
| Mean | $0.060 \pm 0.001$ | $0.091 \pm 0.001$ | $0.116 \pm 0.002$ | $22.52 \pm 0.19$ | $19.25 \pm 0.18$ | $17.67 \pm 0.14$ | $0.48 \pm 0.01$ | $0.51 \pm 0.01$ | $0.49 \pm 0.01$ |

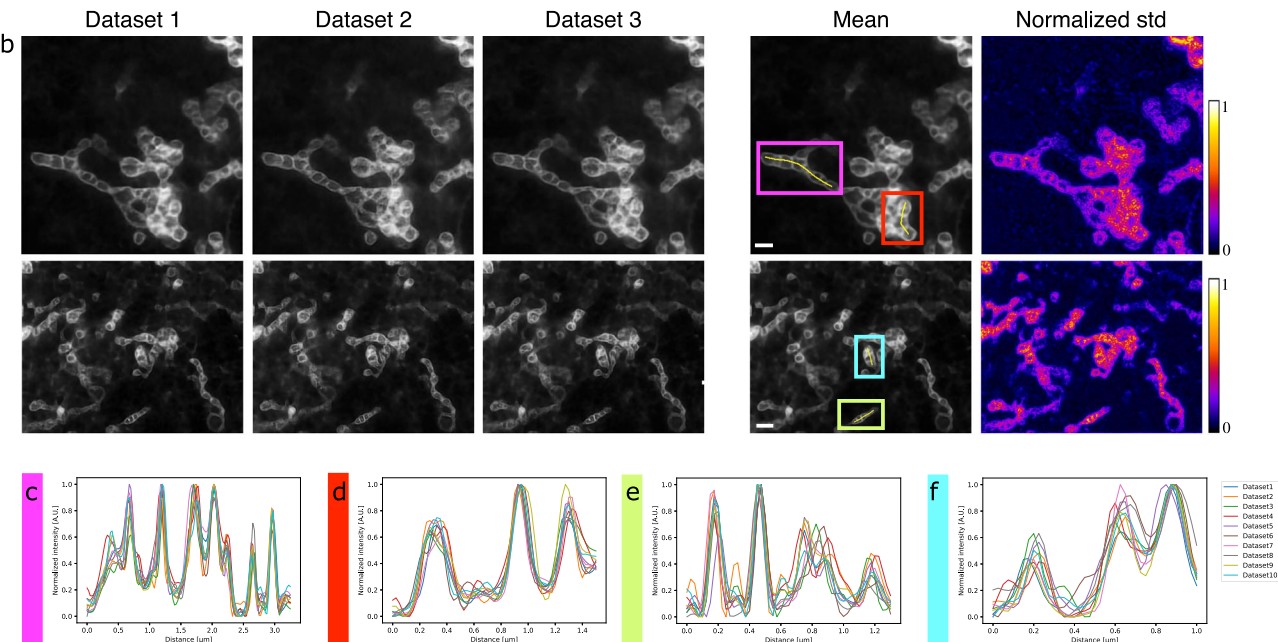

**Fig. 5 | Stability of image restoration by DeepCristae.** Assessment of the stability of DeepCristae by studying the consistency between its predictions obtained with different training. To that end, 10 DeepCristae neural networks were trained with different training data, each one generated with our patch generation method applied to the 24 training images of $D_{synt}$. Note that for this experiment, all networks were initialized with the same weights. **a** Quantitative comparison of the 10 Deep-Cristae models. Metrics were computed on the test set of $D_{synt}$. Data are expressed as mean ± standard deviation. **b** From left to right: predictions of three DeepCristae networks on two images, the average prediction over the 10 trainings and the corresponding pixel-wise normalized standard deviation. Pixel size: 25 nm. Scale bar: 1 µm. **c–f** Comparison of normalized intensity line profiles along a mitochondrion in (**b**) between the 10 trainings. The yellow line, indicated in the corresponding colored inset in (**b**) serves to identify the fluorescence profile.

upper panel, right image and composite zoomed area for comparison) improves the cristae resolution (Raw: 339 ± 248 nm, when measurable; DeepCristae: 94 ± 15 nm) and strongly reduces the variance of paired measurements (Fig. 7d). One of the obvious advantages of LLSM over confocal imaging is to allow continuity between single image planes over large stacks coupled to an extended depth of focus, as illustrated here by the 3D rendering as an oblique projection (Fig. 7c, 3D). Moreover, LLSM is particularly adapted to long range/high frequency imaging on whole living cells, which, coupled to low photon dose illumination, makes it one of the best imaging systems, if not the best, for the highly light-sensitive organelles that are the mitochondria. Applying DeepCristae adds information on cristae ultrastructural organization in the whole mitochondria network of the cell.

Finally, Fast 3D Live-SR and LLSM time series (Fig. 7e, f) were treated for DeepCristae restoration. Cristae ultrastructural features can be observed, while the mitochondrion network undergoes well known dynamic modifications such as fusion or fission processes (Fig. 7e, f, panels of composite zoomed area in both time series; left "RAW " and right "DeepCristae"; Supplementary Movies 1 and 2). Images are of better quality after restoration of Live-SR compared to LLSM images. However, it should be noted the gain in acquisition parameters for the latter in these experiments, with 71 slices per stack and a double channel stack time of 1.3 s versus 14 planes per stack and double channel stack time of 5.6 s for Live-SR. DeepCristae restoration was also tested with an AiryScan 5 LSM 980. It provided similar improvements, although for a 15 planes stack time of about 30 s and with more artifacts appearing after DeepCristae, the nature of which most probably lies in the way the reconstruction of the AiryScan images was carried out from the values determined automatically by the commercial software (Supplementary Fig. 5a–c).

## DeepCristae restoration allows to decipher mitochondria cristae morphodynamics during inter organelles interactions

The most documented membrane-membrane interactions involving mitochondria are the endoplasmic reticulum (ER)–mitochondria contacts, whose functions have been continuously expanded since the 1990s[48,49]. In addition to the ER, mitochondria contact vacuoles/lysosomes, peroxisomes, lipid droplets, endosomes, the Golgi, the plasma membrane and melanosomes[50]. The number of these interactions as well as their duration drastically vary from one type to the other, as they depend on the respective membrane surface of the specific organelles within the cell and their contact time[51]. Their detection may thus require fast and/or long-range 3D imaging. As already mentioned, even high-resolution approaches which are well adapted to decipher ultrastructural features of mitochondria such as cristae, generally fail to capture their dynamic evolution in the 3D space of the whole cell at multiple time scales. This can be critical, if one wants to study inter-organelle membrane interactions and their effects. We next initiate the investigation of endosome/lysosome-mitochondria dynamic interactions by addressing specifically the ultrastructural behavior of the cristae during these contacts. This was done by imaging multiple 3D+time double fluorescence series in Live-SR (represented as a single stack MIP in Fig. 8a, left) or LLSM (represented as a single stack MIP in Supplementary Fig. 5d), where the membranes of the endo-lysosomal pathway were continuously labeled with Plasma Membrane Deep Read (PMDR) (Supplementary Note 1.1). DeepCristae restoration was applied on both datasets. A number of mitochondria dynamic events correlated with endosomal structure behaviors were captured. Only a few of them are here extracted as thumbnail time series (Fig. 8a right and Supplementary Fig. 5e) of zoomed area (colored insets in Fig. 8a left and Supplementary Fig. 5d) from the Live-SR and LLSM acquisitions, respectively. Among others, the formation of endo-

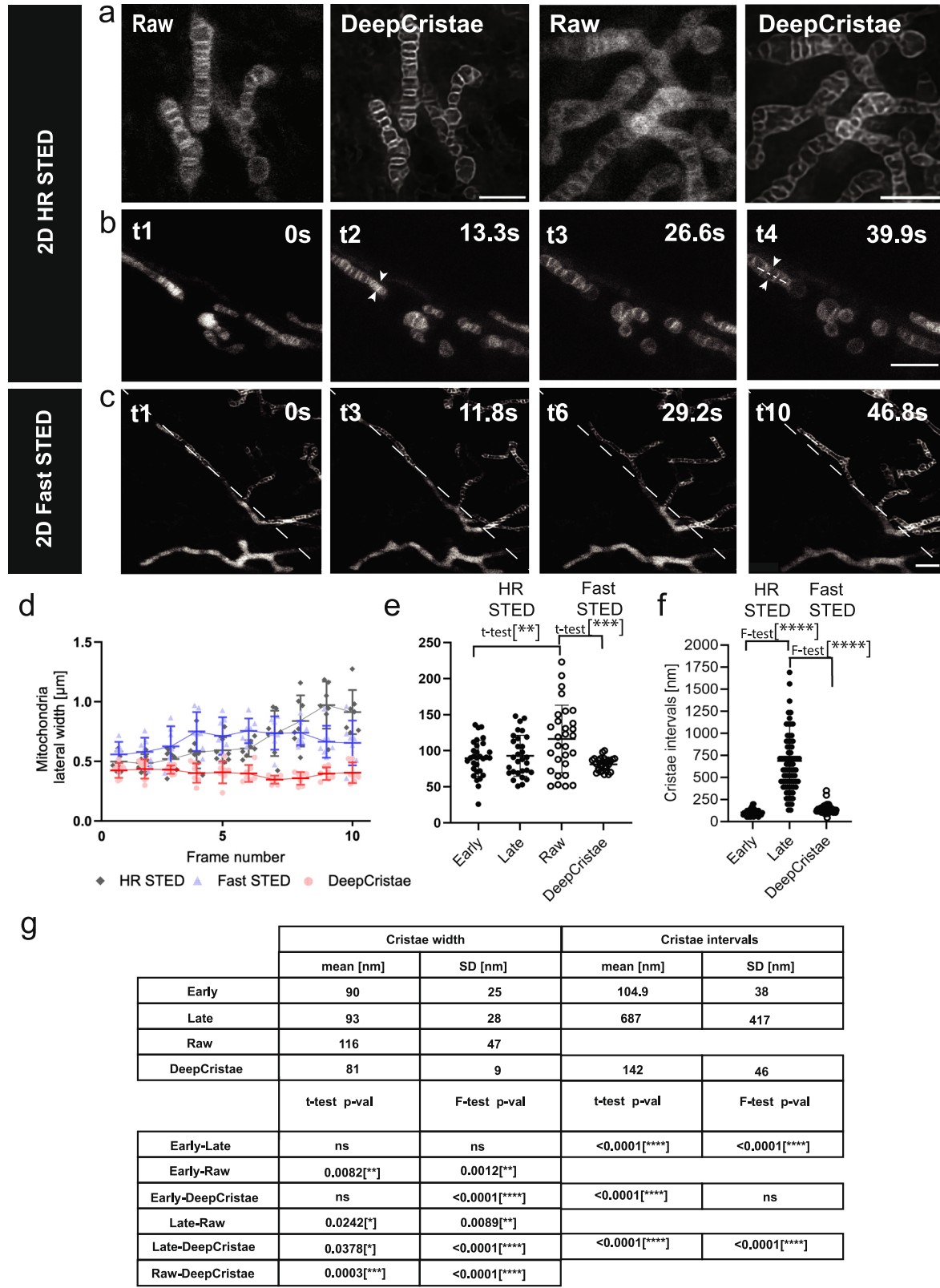

| | Cristae width | | Cristae intervals | |
|---|---|---|---|---|
| | mean [nm] | SD [nm] | mean [nm] | SD [nm] |
| Early | 90 | 25 | 104.9 | 38 |
| Late | 93 | 28 | 687 | 417 |
| Raw | 116 | 47 | | |
| DeepCristae | 81 | 9 | 142 | 46 |
| | t-test p-val | F-test p-val | t-test p-val | F-test p-val |
| Early-Late | ns | ns | <0.0001[****] | <0.0001[****] |
| Early-Raw | 0.0082[**] | 0.0012[**] | | |
| Early-DeepCristae | ns | <0.0001[****] | <0.0001[****] | ns |
| Late-Raw | 0.0242[*] | 0.0089[**] | | |
| Late-DeepCristae | 0.0378[*] | <0.0001[****] | <0.0001[****] | <0.0001[****] |
| Raw-DeepCristae | 0.0003[***] | <0.0001[****] | | |

lysosomes contacts sites with mitochondria (Fig. 8a and Supplementary Movie 3, blue and red insets), very long confinement of endo-lysosomes within the mitochondria network (Fig. 8a and Supplementary Movie 3, orange inset) and image series of endo-lysosomes appearing to pull a small mitochondrion from one to another elongated tubules of mitochondria (Fig. 8a, red inset). DeepCristae restoration on the space-time localization of

these events can also be evaluated dynamically (Supplementary Movie 3). Similar events are followed with LLSM, such as the fission of mitochondria at a contact site with an endo/lysosome vesicle (Supplementary Fig. 5e, orange inset) and long confinement of an endo/lysosome vesicle within the mitochondria network (Supplementary Fig. 5d and Supplementary Movie 4, green inset). The main advantage of the LLS modality (fast frame

**Fig. 6 | DeepCristae reveals mitochondria cristae from low resolution 2D live STED. a–c** 2D live STED imaging of RPE1 cells labeled with PKMITO-Orange. **a** Single time points of 2D high-resolution (HR) STED (Raw) imaging (see Supplementary Note 1) before and after DeepCristae restoration. **b** First 4 time points of a 10-image time series ($\Delta t \sim 13s$) using 2D HR STED. Phototoxic damage is shown by the swelling of the mitochondria. Pixel size is $50 \times 50$ nm in (**a**) and $25 \times 25$ nm in (**b**). **c** Time series of 2D Fast STED, reducing the time delay between time points ($\Delta t \sim 5.9s$) and minimizing mitochondrial damage. Each thumbnail is diagonally divided into raw 2D Fast low-resolution (LR) STED images (bottom-left) and after DeepCristae restoration (top-right). Note: a rescaling factor of 1.87 was applied to the LR image data before DeepCristae inference to match the training mitochondria settings (see Results, "Robustness of DeepCristae with respect to noise, blur, and mitochondria scale in low-resolution images"). Scale Bars in (**a–c**) are: 2 µm. **d** Lateral widths of 7 mitochondria measured at each time point (series of 10 time points) in HR STED and Fast (LR) STED, before and after DeepCristae restoration

(line profiles as indicated by arrowheads in (**b**) t2 and t4). Line profiles were fitted to a Gaussian model and the Full Width Half Maximum (FWHM) was measured, as described in Supplementary Note 1. Data are presented as mean ± SD. **e** Cristae widths were measured as described in (**d**) for 30 cristae from 20 mitochondria (from 4 distinct image series) at early and late time points of HR STED, and for 31 cristae of Fast (LR) STED, before (Raw; two measurements outside the ordinate scale) and after DeepCristae restoration (line profiles as illustrated in (**b**) t4 and (**c**) t10). Data are expressed as mean ± SD. Student's $t$-test: ** ($p$-value = 0.0082), ***($p$-value = 0.0003). **f** Distances between two cristae (cristae intervals) measured from peak-to-peak intensity in plot profiles (measurements from Raw Fast STED imaging was not possible). Early: $N = 56$ from 2 series; Late: $N = 80$ from 2 series; DeepCristae: $N = 60$ from 3 series, distributed in all time points. Data are expressed as mean ± SD. $F$-test: ****($p$-value < 0.0001). **g** Statistics table, including significance testing with Student's $t$-test and Fisher's test; ns non-significant.

rate, low photon illumination of the sample coupled to whole cell 3D acquisition) is the improvement of the time resolution of the data series (or long-range acquisition). Consequently, fast events involving endo-lysosome contacts with mitochondria are easier to capture and these dynamics are precisely deciphered. For instance, one may extract first (Supplementary Fig. 5d, blue inset), probably a fusion process (Supplementary Fig. 5e, blue inset, from time point 105 to time point 113; $\Delta t = 8s$), and second a fission process (Supplementary Movie 4, from time point 192 to time point 198; $\Delta t = 6s$). At each time point, the DeepCristae restored mitochondria and denoised/deconvoluted endo-lysosomes double-labeled images (Supplementary Note 1) are paired to non-treated images (right and left panel, respectively, of thumbnails time series in Fig. 8a and Supplementary Fig. 5e). While cristae resolution in LLSM does not reach that obtained with Live-SR, DeepCristae restoration brings values closer together (Fig. 7b, d).

In all situations and for both intermediate HR (Live-SR) and diffracted limited (LLSM) (Supplementary Fig. 5d) imaging modalities, DeepCristae restoration provides ultrastructural information on the positioning, density, and dynamics of mitochondria cristae. We then wanted to quantitatively assess how the dynamic architecture of the mitochondria internal membrane during endo/lysosomes-mitochondria interaction could be revealed with DeepCristae. We focused on the fission process. To do so, we first selected 21 distinct 3D+time image series from the Live-SR datasets, in which mitochondria fission was monitored. Intensity line plots were measured along mitochondria on some time points framing the fission event (Fig. 8b). This was done on both DeepCristae-restored and unrestored individual time points in a "blind" manner, meaning without looking in the second channel depicting the location of endo-lysosomes. Measurements of "peak-to-peak" intervals between cristae, were only possible in the DeepCristae restored images and show an increased density after fission occurs (Fig. 8c, dark circles). Interestingly, by overlaying the second channel in a second step, 62% (13 out of 21) of these selected time series showed proximity if not direct contact between endo/lysosomes and mitochondria at the site where mitochondria fission is observed (Fig. 8c, red circles; Fig. 8d for statistics). Similarly, 32 distinct 3D+time image series from Live-SR datasets of labeled mitochondria (PKMITO-Orange) and lysosomes (SIR_lysosome) were analyzed (Supplementary Fig. 6). In this case, 59% (19 out of 32) of the selected time series showed proximity between lysosomes and mitochondria, where mitochondrial fission was observed (Supplementary Fig. 6b, c). While still preliminary and not deciphering the exact nature of the endosomal compartments involved (i.e., PMDR labels the overall endo-lysosomal pathway), this illustrates how DeepCristae would represent an asset to quantitatively study the dynamic architecture of the mitochondria internal membrane during diverse dynamic processes or in particular physiological or constrained conditions.

## Discussion

Mitochondrial membrane architecture is essential for the many functions of mitochondria. In particular, mitochondria cristae are the main site of energy production and are dynamic ultrastructures that remodel

in response to various cellular stimuli and natural processes (apoptosis[1]; aging[52]). Therefore, understanding the structure and dynamics of cristae is vital for comprehending mitochondrial function and its implications in cellular physiology and diseases. High-resolution microscopy coupled with robust mitochondrial probes[7,9] are key recent developments that started to reveal the fine details of mitochondrial cristae structure and organization, overcoming the limitations of conventional microscopy. However, imaging at high spatial and temporal resolution remains a challenge.

DeepCristae exploits the power of deep learning to reveal cristae in images taken with low photon illumination, enabling clearer visualization and analysis of mitochondria cristae in living cells without interfering with the natural behavior of mitochondria. While it has been trained on a dedicated dataset that was created from real high-resolution 2D STED images, we have shown that it operates for a wide range of optical resolutions, from diffraction-limited to intermediate high-resolution microscopy, providing researchers with a powerful tool to study cristae dynamics without compromising their structural integrity or functionality.

While there are other deep learning approaches available for revealing cristae ultrastructure[21,22,39,41], DeepCristae offers unprecedented advantages. First, thanks to a well-defined training loss dedicated to the restoration of mitochondria signals; it outperforms state-of-the-art methods. Secondly, it not only makes it possible to visualize and restore cristae dynamics in 2D STED nanoscopy with minimal illumination and without damaging mitochondria but more importantly, it extends these capabilities to other high-resolution imaging techniques such as Live-SR and AiryScan, more suited to such 3D dynamics. Finally, DeepCristae can be applied to advanced microscopy techniques such as LLSM, enabling fast and long-duration 3D+time acquisitions within the diffraction-limited range. This versatility makes DeepCristae a unique and valuable solution for studying cristae dynamics across a range of spatial and temporal scales.

Overall, our results show that fluorescence microscopy combined with DeepCristae enables long-term/fast dynamic observation of cristae behavior and organization with high quality. To illustrate the contribution of our approach to biological phenomena that are likely to involve the functional structure of mitochondria, we have chosen to focus on inter-organelle interactions and their consequences. While mitochondria-associated ER membranes, the biochemical composition of the contact sites and diverse physiological and disease-related functions have been extensively studied over the decades[53,54], it is increasingly recognized that other organelle contacts have a vital role in diverse cellular functions[55]. More recently, there has been growing interest in quantifying other membrane interactions with mitochondria and their cell distribution in space and time[51], in particular within the endo-lysosomal pathway and their contribution to the fission/fusion process of the mitochondria network[56]. Here, while confirming the coincidence of contacts between the endo-lysosomal membrane and mitochondria, we enlightened the change of cristae density during fission (Fig. 8 and Supplementary Fig. 6). This density as well as complex cristae arrangements depends on cell types and metabolic activities[4,57], not talking

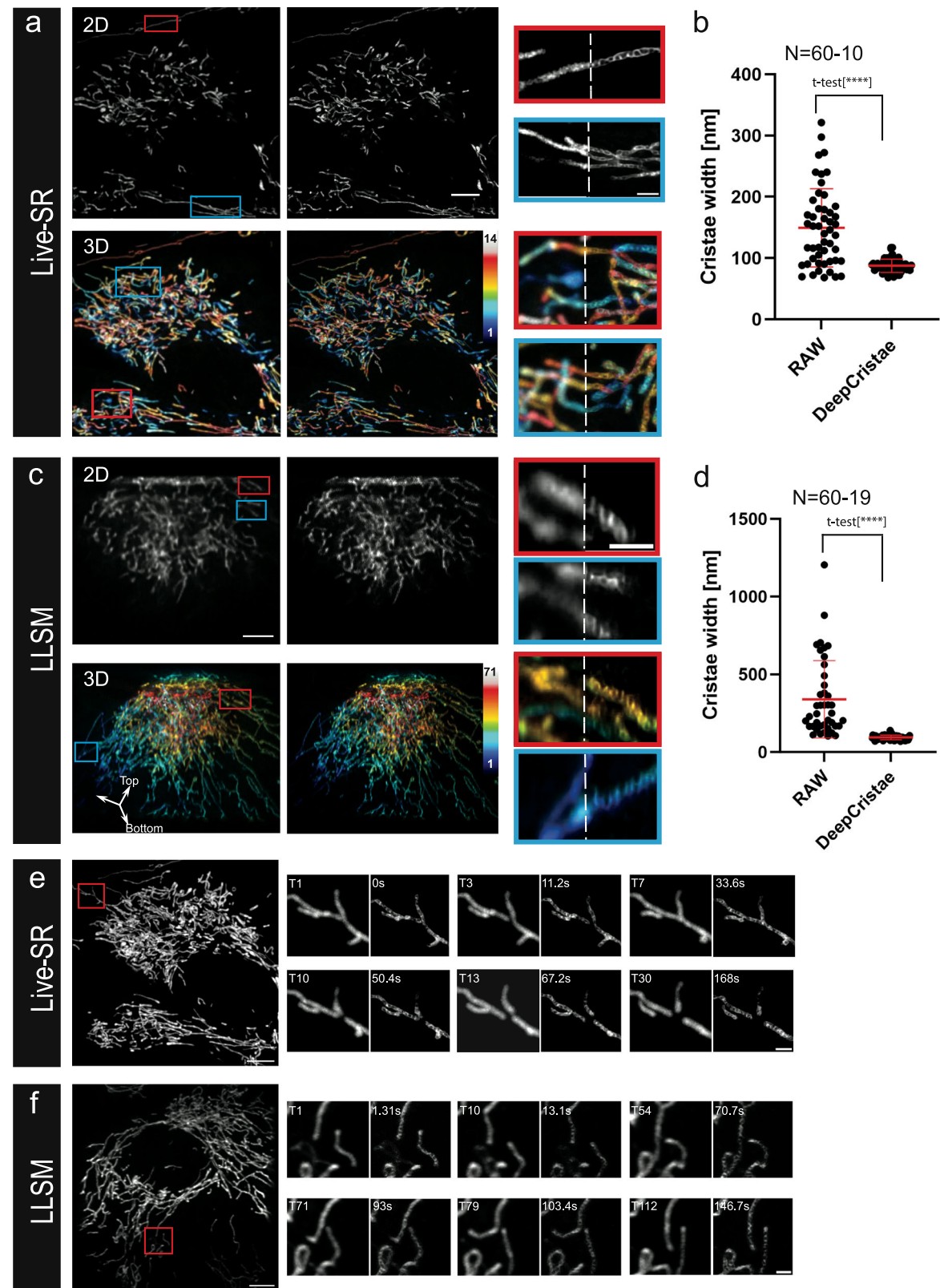

of obvious modifications induced by environmental conditions. Until now, to provide a dynamic view of individuals and groups of cristae required 3D nanoscopy or linear SIM[16], which are not always compatible with the time frame required to capture the event of interest. In this respect, DeepCristae might be an asset to compare the cristae dynamics in different cell types and in these various conditions.

However, as with any image restoration method, scientists may be concerned by the reliability of DeepCristae to accurately restore mitochondria cristae and not hallucinate them. This is why we investigated the robustness, stability and limits of our method (Figs. 4 and 5, Supplementary Figs. 2, 3 and 4). We worked out different conditions of use to be respected to guarantee good quality and truthfulness of the results. It is important to feed

**Fig. 7 | DeepCristae restoration enhances cristae width resolution in 3D and 3D +live imaging. a** 2D plane (top) and 3D MIP (Maximum Intensity Projection of 14 planes) (bottom) of an RPE1 cell labeled with PKMITO-Orange, acquired using an SD microscope with a Live-SR module, before (left) and after DeepCristae (right) restoration. Thumbnails are zoomed areas corresponding to the insets (red and blue) and are composites of RAW and DeepCristae images. Color scale bar indicates mitochondria position ($z$-step: 200 nm) from bottom to top (bottom right). **b** Cristae widths were measured as in Fig. 6e; each individual measurement in DeepCristae restored images is compared to its equivalent in RAW images, except for 10 cristae that were not measurable in RAW ($N = 60$ and $N = 60$-10, respectively). Data are expressed as mean ± SD (DeepCristae: 87 ± 11 nm; RAW: 149 ± 64 nm; Student's $t$-test, [****] $p < 0.0001$). **c** One section plane (top) and 3D reconstructed MIP of 71 planes (bottom) of a RPE1 cell labeled with PKMITO-Orange, acquired with a Lattice Light Sheet Microscope (LLSM) in dithered mode (after realignment (deskew) and a Richardson-Lucy deconvolution), before (left, RAW) and after DeepCristae (right). Thumbnails are zoomed areas corresponding to the insets (red and blue) and are composites of RAW and DeepCristae restored images. Color scale bar indicates mitochondria position ($z$-step: 325 nm) from bottom to top (bottom right). **d** Cristae widths were measured as in Fig. 6e in DeepCristae restored images and compared to their RAW equivalents, when possible ($N = 60$ and $N = 60$-19, respectively). Data are expressed as mean ± SD (DeepCristae: 94 ± 15 nm; RAW: 339 ± 248 nm; Student's $t$-test, [****] $p < 0.0001$). **e, f** 3D+time imaging using Live-SR (**e**) or LLSM (**f**). MIP of single time points are shown (left images). Insets indicated in red are zoomed in the thumbnails (right image series) to illustrate fusion or fission dynamics of mitochondria. The selected zoomed areas are shown at different time points before (left panel) and after (right panel) DeepCristae restoration. Time frames between stacks are 5.6 s and 1.31 s in double-channel acquisition for Live-SR and LLSM, respectively. Scale bars are 5 μm in full field images and 1 μm in zoomed thumbnails. Before DeepCristae restoration, rescaling factors of 2.6 and 4.16 were applied to each raw Live-SR and LLSM dataset, respectively (see Results, "Robustness of DeepCristae with respect to noise, blur, and mitochondria scale in low-resolution images").

---

DeepCristae with images containing mitochondria whose average width in pixels is close to the one seen during the training. Concerning the microscope settings, it is better to ensure that the level of noise and blurring in the input images is equivalent to or better than the one present in the training data (which was quite high in our training). Under these conditions of use, across all our experiments on real data and through different microscopy modalities, no hallucination was observed: a consistency between line profiles along mitochondria between raw and restored data was always observed (Figs. 6 and 7).

Like cytoskeletal elements, the mitochondrial ultrastructure is a key element for comparing the performance of new super-resolution microscopy techniques. In terms of applications, DeepCristae makes it possible to track the evolution of mitochondrial cristae morphology over time, during interactions with other membrane components of the cell, or under extracellular conditions that mimic various pathological or stress situations.

## Methods

In this section, we present the main features of DeepCristae. We first present the dataset we created from real 2D STED images to train and evaluate the network. Then, we overview our network architecture and present the novel learning loss function, which prioritizes the restoration of specific pixels. We finally detail the image patch-sampling method used during the training step to efficiently increase the size of our training set and thus improve the learning process.

### Generation of the 2D STED dataset - $D_{synt}$

As mitochondria are living organelles, mostly organized as a quite fast-moving network in RPE1 cells (Supplementary Note 1.1), the acquisition of a pair of high and low-resolution images at the exact same time point is impossible. To train and quantitatively validate DeepCristae, we thus created a dataset (Fig. 1a–c), called $D_{synt}$, from 33 acquired 2D HR STED images (25 × 25 nm) that we denote $I_{HR}$. More information on the acquisition of the images $I_{HR}$ are available in Supplementary Note 1.2.1.

First, we degraded the images $I_{HR}$ to obtain LR images of mitochondria, denoted $I_{synt_{LR}}$ (Fig. 1b), that will serve as input to the neural network. To that end, we first applied a Gaussian filter of standard deviation $\sigma_{blur} = 3.25$ pixels to the images $I_{HR}$ in order to approximate the blurring effect due to the point spread function of the microscope. Then, we added a Poisson-Gaussian noise ($\sigma_{noise} = 4.0$), consistently with noises observed in real STED images. The parameters $\sigma_{blur}$ and $\sigma_{noise}$ were set to create pertinent input data that mimic real LR STED images (Supplementary Note 1.2.1). Note that the value of $\sigma_{noise}$ was chosen in line with our data where the maximum intensity varies between 56 and 356. Second, we paired the LR STED images $I_{synt_{LR}}$) with their restored counterpart, the HR STED images $I_{HR}$ that are considered as ground truths. Finally, we split the dataset $D_{synt}$ into 24 training images and 9 test images. Note that to improve the training, we enhanced the mitochondria cristae in the images $I_{HR}$ of the training set

using the Richardson-Lucy algorithm[30,31] (Fig. 1c). Other non-iterative deconvolution algorithms were tried, such as SPITFIR(e)[33] or Wiener[32], but the results obtained after training were not as good.

To further increase the size of the training set, data augmentation (Fig. 1d) and patch sampling (Fig. 1e and described in Methods "Image patch sampling for the training step") are performed on the pair of LR images $I_{synt_{LR}}$ and HR STED images $I_{HR}$. The dataset is first augmented by applying three different rotations to the images (90°, 180°, and 270°). Then, a shrink transform, and horizontal and vertical flips are successively applied to 25% of the augmented dataset, randomly selected. The final training set is made of 1824 patches of size 128 × 128 pixels, whose 20% are used for the validation set and so that there is no overlap with the patches used for training (summary in Supplementary Table 1).

The 9 HR STED test images $I_{HR}$ have different levels of noise and blur due to out-of-focus light mitochondria. For the evaluation of our method, we selected 26 ROIs out of these 9 $I_{HR}$ images where the mitochondria are in the focal plane, that we have labeled as "test images".

### Network architecture

We used the network proposed by Weigert et al.[19] as the backbone of the CNN architecture, itself built upon the U-Net[58]. It has a contracting path and an expansive path, each one consisting of 3 sequential downsampling and upsampling blocks, respectively. Each block of the first path is skip-connected to the associated one of the expansive paths. The contracting path consists of two successive 3 × 3 convolutions, each followed by a Rectified Linear Unit (ReLU), and a 2 × 2 max pooling operation with stride 2 for downsampling. Every depth in the expansive path consists of a 2 × 2 up-sampling of the feature map, concatenated with the corresponding feature map from the contracting path, followed by two 3 × 3 convolutions with a ReLU activation function. At the final layer, one 1 × 1 convolution is used. The output results from an additive assembly between the input of the neural network and the last layer's output. The network (Fig. 1f) outputs the same size restored images. Note that the network was trained with patches of size 128 × 128 pixels, but in the inference step (Fig. 1g–i), raw data of any size can be used as input.

### Design of the training loss

We present our new loss, the Similarity Component Prioritization (*SCoP*) loss, that has been designed to better restore mitochondria cristae. Most losses and metrics used to train networks or to evaluate the quality of restorations compute the score on the whole image, giving the same weight to any pixel. For example, the *MAE* computes the mean absolute error between the prediction and the target image, while the *SSIM*, despite not basing the calculation on pixels-to-pixels difference, computes the similarity among all the pixels of both images. Instead, our purpose is to focus on informative pixels corresponding to target structures in images. Indeed, the dark and noisy background occupies most of fluorescence images,

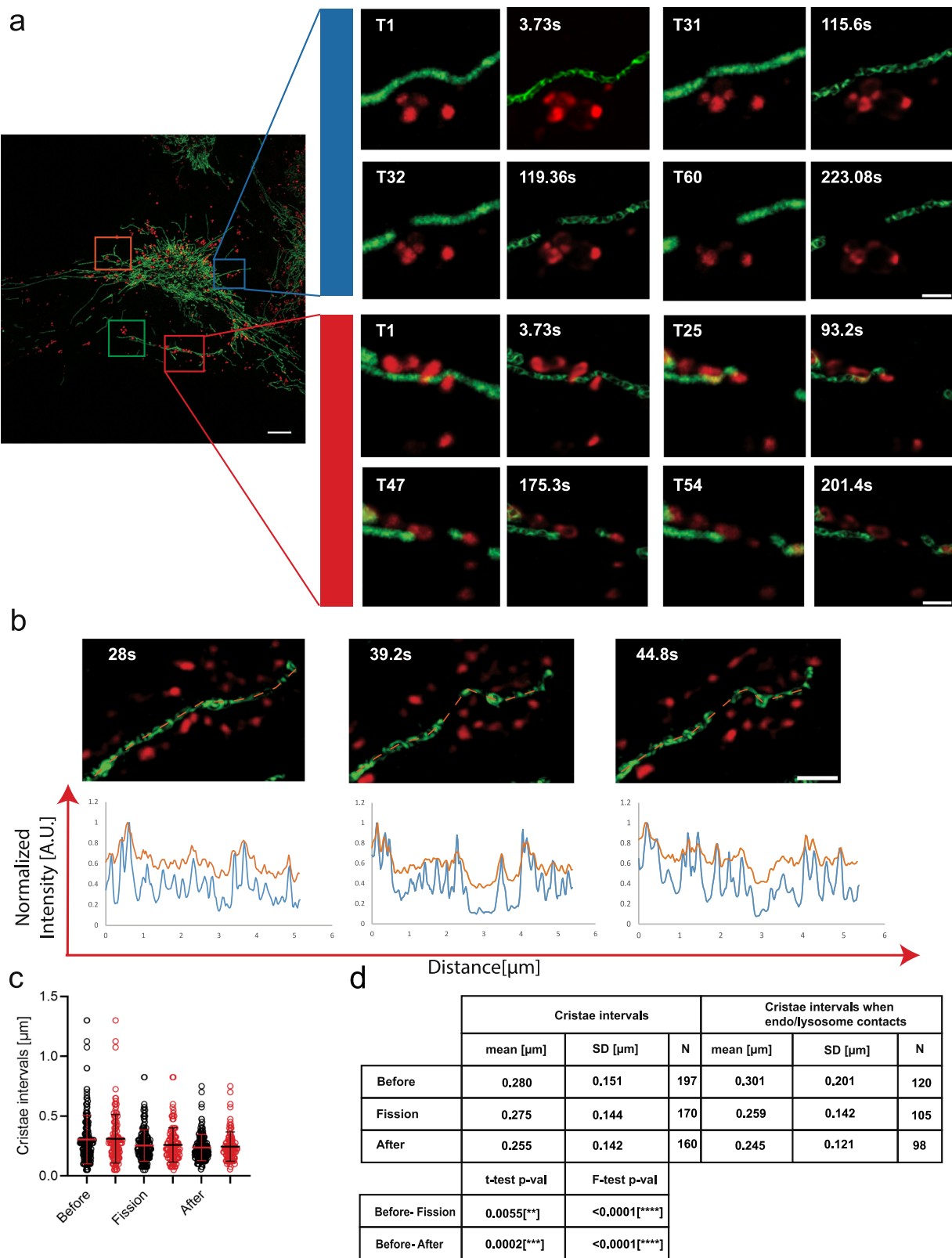

which alters the learning. To overcome this issue, we introduce *SCoP*, a novel loss which adaptively prioritizes the restoration of mitochondria pixels.

Our *SCoP* loss is built upon the structural dissimilarity (*DSSIM*) measure. Consider $(i, j)$ the spatial coordinates of a given pixel and a patch of size $(N, N)$. The loss formula between a target image $y$ and its prediction $\hat{y}$ is given by

$$SCoP(y, \hat{y}) = \frac{1}{N^2} \sum_{i=1}^{N} \sum_{j=1}^{N} \left( \frac{1 - SSIM_{y,\hat{y}}^{map}(i,j)}{2} \right)^{\gamma_{i,j}} \quad (1)$$

**Fig. 8 | DeepCristae reveals 3D+time cristae morphology during endo/lysosome mitochondria interactions. a, b** RPE1 cells incubated for 4 h with Cell Mask Plasma Membrane (PM) Deep Red (red) were labeled with PKMITO-Orange (green) in the last 15 min. (**a-left**) A Maximum Intensity Projection (MIP) (20 planes; stack time = 1.86 s/channel, time point T1 out of 60) image acquired with Live-SR microscopy is shown after DeepCristae restoration of the mitochondria (green channel), as well as after denoising (ND-SAFIR) and Richardson-Lucy (RL) deconvolution of the endo/lysosomes (red channel). Colored Insets indicate intra-cellular locations with dynamic events of interest. (**a-right**) Thumbnails show selected time points of blue and red zoomed areas, as indicated by insets. They are presented as paired images: before (left panels) and after (right panels) DeepCristae restoration. Both time points (left panels) and time frames in seconds (right panels) are indicated for comparison. Full acquisition video (T1–T60), including the four regions, is provided as Supplementary Movie 3. Scale bars are 5 μm in the full field image and 1 μm in zoomed thumbnails. **b** Thumbnails show selected time points of a zoomed area, from a different cell than (**a**), as MIP (14 planes; stack time = 2.8 s/channel, time points T6, T8 and T9) from an image acquired with Live-SR microscopy after DeepCristae restoration (green channel), as well as after denoising (ND-SAFIR) and RL deconvolution for endo/lysosomes labeling (red channel). It represents a location where mitochondria fission is occurring. Profile lines are indicated in orange in (**b**) top. The bottom plots illustrate the PKMITO-Orange intensity line plot before (orange line) and after DeepCristae restoration (blue line) at the same time points. Scale bar in (**b**) is 1 μm. **c** Graphs measuring the "peak-to-peak" intervals between cristae in DeepCristae restored images, measured before, during and after fission. Measurements were first taken from 21 distinct time series, in a blinded manner (black circles; number of peak-to-peak intervals (N) are indicated in (**d**)), and then where endo/lysosomes contacts with mitochondria occur in the same series (13 of 21 distinct time series; red circles). Error bars indicate mean ± standard deviation (SD). **d** Statistics table for cristae intervals measurements, including mean, standard deviation (SD), and significance levels using Student's and Fisher's tests. Note that a rescaling factor of 2.6 was applied to raw Live-SR data prior to DeepCristae restoration. This is done to comply with the usage conditions of DeepCristae (see Results, "Robustness of DeepCristae with respect to noise, blur and mitochondria scale in the low-resolution images").

where $SSIM_{y,\hat{y}}^{map}$ is the map of the local structural similarity ($SSIM$) values for corresponding pixels between the images $y$ and $\hat{y}$. Each $SSIM$ value ranges in $[-1, 1]$, where $-1$ (1, respectively) testifies of a bad (very good similarity, respectively) between $y(i,j)$ and $\hat{y}(i,j)$. The parameter $\gamma_{i,j}$ prioritizes the restoration of specific regions of interest. In our case, we chose $\gamma_{i,j} = 1$ if the pixel $(i,j)$ belongs to a mitochondrion, 4 otherwise. In this way, we encourage the network to focus on restoring mitochondria pixels and reduce the influence of a poorly restored background on the loss. Determining whether a pixel belongs to the background or to a mitochondrion can be performed automatically (using our method described in Methods "Image patch sampling for the training step - Thresholding") or manually by using any binary segmentation provided by the user.

### Data normalization

Our training images of $D_{synt}$ have different ranges of intensity values. To homogenize them, we normalized the input data and their corresponding ground truth to a common distribution of intensity values with the percentile normalizer. This normalization also has the advantage of excluding outliers, which are very frequent in microscopy imaging due to noise and luminance. The percentile normalization of an image $I$ is defined as

$$I_{norm} = \frac{I - perc(I, p_{low})}{perc(I, p_{high}) - perc(I, p_{low})} \qquad (2)$$

where $perc(I, p)$ is the $p$-th percentile of $I$. We used $p_{low} = 2$ and $p_{high} = 99.8$. This step is also performed during the inference step on any input data.

### Image patch sampling for the training step

Our model is trained on the training set of $D_{synt}$ containing 24 images (96 after data augmentation) of different sizes. In order to homogenize and increase the training dataset, we performed patch sampling. We sampled each input training image $I \in R^{W \times L}$, defined over the grid $\Omega$ of size $W \times L$, within $N_I = \left[\frac{W}{128}\right] * \left[\frac{L}{128}\right]$ patches of size $128 \times 128$. As our images contain more background pixels than mitochondria pixels, grid or simple random patch sampling will end in too many empty patches. This can degrade the training of our model. Instead, we perform a random sampling focusing on the regions of interest, the mitochondria. Our pipeline (Supplementary Fig. 7) is described as follows.

- *Anscombe transform.* To detect the areas of interest, we need to enhance the mitochondria signal with respect to the noise. To do this, we first remove the Poisson-Gaussian noise in STED images. This is achieved by applying an Anscombe transform, which enables to stabilize noise variance and to approximately convert Poisson-Gaussian noise into white Gaussian noise with a constant variance.

The Anscombe transform of an image $I$ is given by

$$I_{Ansc}(i,j) = 2\sqrt{\frac{3}{8} + I(i,j)}, \forall (i,j) \in \Omega \qquad (3)$$

- *Z-score.* Then, we compute the Z-score map defined as

$$Z(i,j) = \frac{I_{Ansc}(i,j) - \hat{\mu}_\epsilon}{\hat{\sigma}_\epsilon}, \forall (i,j) \in \Omega \qquad (4)$$

where $\hat{\mu}_\epsilon$ and $\hat{\sigma}_\epsilon$ are the estimated mean and standard deviation of the Gaussian noise $\epsilon$, respectively. Since most of the pixels in $I_{Ansc}$ belong to the background, we consider $\hat{\mu}_\epsilon = median(\{I_{Ansc}(i,j)\}_{(i,j)\epsilon\Omega})$. For $\hat{\sigma}_\epsilon$, we use a robust estimator derived from the Median Absolute Deviation ($MAD$) such that $\hat{\sigma}_\epsilon = 1.4826 \cdot median(\{|r(i,j)|\}_{(i,j)\epsilon\Omega})$, where $r(i,j) = \frac{2I_{Ansc}(i,j) - I_{Ansc}(i+1,j) - I_{Ansc}(i,j+1)}{\sqrt{6}}, \forall(i,j)\epsilon\Omega$, are the pseudo-residuals. In fact, under the hypothesis of having a white Gaussian noise and that the noise-free image is piecewise smooth in a local neighborhood, we have that $\hat{\sigma}_\epsilon^2 = E[r^2(i,j)]$.

- *Thresholding.* The higher the Z-score in Eq. (4), the higher the pixel value is above the mean of the measured noise and therefore the pixel $(i,j)$ is considered as a pixel of interest. We apply a threshold $c$, in a way that any pixel $(i,j) \epsilon \Omega$ such that $Z(i,j) > c$ is considered as a mito-chondria pixel. We denote this set as $\Omega_{mito}$. The threshold is auto-matically adapted for each training image. Starting from a fixed high value of 30, while $\Omega_{mito}$ does not contain a minimum of 10% mito-chondrial information (i.e., $\#\Omega_{mito} < 10\% \#\Omega$, where $\#\Omega$ and $\#\Omega_{mito}$ denote the number of pixels in the sets $\Omega$ and $\Omega_{mito}$, respectively), we subtract 5 from the threshold value. This creates a binary mask on which we apply a median to remove the surrounding noise. This automatic procedure avoids cumbersome manual annotations. Note that this mask can also be used to compute the parameter $\gamma_{i,j}$ in our loss (see Eq. (1)).

- *ROIs selection.* From $\Omega_{mito}$, we randomly choose $N_I$ different pixels to be the center of ROIs of size $128 \times 128$ pixels. Thus, the more pixels of mitochondria a ROI contains, the more likely it is to be chosen. The following conditions have to be respected: (i) the ROI centers should not belong to the borders of the image; (ii) to avoid redundancy, a minimum distance of 60 pixels is established between each pairwise ROI center. The resulting ROIs are finally used to create the patches from the normalized training data (see above "Data normalization").

## Network evaluation

In addition to a quantitative comparison to the state-of-the-art methods and experiments to show the reliability of our method (Results), an ablation study was also performed (Supplementary Fig. 8 and Supplementary Note 2.3) to highlight the individual contribution of key components of our method. More details about the evaluation metrics and the implementation details of DeepCristae are also given in Supplementary Note 2.1 and Supplementary Note 2.2.1, respectively.

## Other methods and materials

Cell culture and biological materials, fluorescence labeling, all used microscopy techniques, image acquisition protocols and quantitative measurements are detailed in the Supplementary Note 1. PKMITO dyes are commercially available at Spirochrome (Stein-am-Rhein, Switzerland) and Genvivo Biotech (Nanjing, China). The hTERT-immortalized RPE1 cells (Human Retinal Pigment Epithelial Cell) were purchased from ATCC (CRL-4000).

## Statistics and reproducibility

Student $t$-test and Fisher-test were performed using GraphPad Prism 9 and Excel Microsoft 365. We performed unpaired student $t$-test with two-sided $p$-values. In all cases, $p$-values more than 0.05 were considered not significant. Sample sizes were not predetermined using statistical methods. The sample sizes are indicated in the figure legends, and error bars represent the standard deviations. The number and definition of experimental replicates are provided in the figure legends, where applicable.

## Reporting summary

Further information on research design is available in the Nature Portfolio Reporting Summary linked to this article.

## Data availability

Data generated during this study are available in Figshare[59] with the identifier https://doi.org/10.6084/m9.figshare.26940892.

## Code availability

DeepCristae source code used in this publication is open-source and published under the BSD 3-Clause "Original" or "Old" License. It is available at https://gitlab.inria.fr/anbadoua/DeepCristae. It can also be found in Zenodo[60] with the identifier https://doi.org/10.5281/zenodo.14714199.

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

## Acknowledgements

This work was supported by the France-BioImaging Infrastructure (French National Research Agency, ANR-10-INBS-04-07, "Investments for the future") and the Labex Cell(n)Scale (ANR-11-LABX-0038) as part of the Idex PSL (ANR-10-IDEX-0001-02). We acknowledge the Cell and Tissue Imaging (PICT IBiSA, Institut Curie) and the IMACHEM (Collège de France) platforms, also members of the national infrastructure France-BioImaging (ANR-10-INBS-04-01) for access to and maintaining the spinning-disk, Airyscan and STED microscopes. We also wish to thank M. Maurin from Inserm U932 for his help in a preliminary study on STED image acquisition.

## Author contributions

A.B., L.L., C.-A.V.-C., and J.S. conceived the project. S.P. and A.B. designed the framework of DeepCristae, and conducted benchmarks and every experiment relative to the reliability of the network. S.P. implemented the code of DeepCristae and the Jupyter notebooks. L.L., J.S., and C.-A.V.-C. designed the biological experiments. L.L. and J.S. prepared samples. L.L. with the aid of J.D. performed the acquisitions. L.L. and C.-A.V.-C. analyzed and organized the biological data. L.L., S.P., J.S., and A.B. prepared figures and videos. T.L. and Z.C. provided the mitochondrial dye. A.B., S.P., J.S., L.L., and C.-A.V.-C. wrote the manuscript. All authors critically discussed the results and commented on the manuscript. A.B., J.S., and C.K. supervised the research.

## Competing interests

The authors declare the following competing interest: Z.C. is an inventor of the patent on the mitochondria dye described in this work. All other authors have no competing interests to declare.
