## [Transparent Peer Review file · Communications Biology]

DeepCristae, a CNN for the restoration of mitochondria cristae in live microscopy images

Corresponding Author: Dr Anaïs Badoual

Version 0:

Reviewer comments:

Reviewer #1

(Remarks to the Author)

These authors have developed a deep-learning method to restore mitochondrial cristae from low-resolution microscopy images. Their current method demonstrated several critical advantages over the previous methods.1) They trained their network by designing a training loss that emphasized similarity on mitochondria but not the background.2) They also show the superiority of their method over existing methods.3) Most importantly, they are among the first to address the possible hallucinations induced by the DL method, which I believe shall be encouraged. Overall, I support the publication of this paper in Communications Biology, provided that they address a few minor points (mentioned below) in the revision.

Minor:

1. In Fig.3, the authors have shown representative examples of DL-assisted cristae reconstruction against GT. The image qualities of 3b and 3C were quite impressive, while the 3d and 3e were marginal due to the poor quality of the GT. I wonder whether the authors may design metrics to describe quantitative information regarding the cristae, such as their numbers and sizes. If that could be done, it could be extended to be a more meaningful criterion for evaluating mitochondria.
2. Suppl Fig. 3 tested the reliability of image restoration by DeepCristae according to the level of noise and blur in the image. This is quite important, so I suggest it be incorporated into the main text, not supplementary.
3. The authors stated, "For each ROI, a comparison of normalized intensity profiles between the input LR image, DeepCristae restored image and the ground truth image is performed. A consistency between the cristae restored by DeepCristae and the ones present in the ground truths is observed overall, and no meaningful "hallucination" is observed." This indeed touches on a very important point. I would better understand and appreciate the work if the authors could spend more time elaborating on this point.

Reviewer #2

(Remarks to the Author)

In this paper, the authors have trained a deep learning super-resolution model specifically for mitochondrial cristae by innovating (1) a novel loss function and (2) patch sampling techniques. They have also enhanced existing image-level metrics to effectively evaluate the super-resolution quality of mitochondria and cristae, reporting that their model, named DeepCristae, outperforms existing methods. DeepCristae was applied to data obtained through various imaging techniques including live imaging on cristae dynamics and Endosome/Lysosome-Mitochondria contacts. While the live imaging results presented in the manuscript are intriguing, and their method appears promising for future analyses of ultrastructural dynamics, there are still several important unresolved issues in evaluating the strengths of their approach.

Major Points

1. The order in which the experiments are described is confusing. In Figure 2, DeepCristae is applied to Dsynt, yet the manuscript seems to suggest that Figure 3 is the first figure demonstrating the application of DeepCristae to Dsynt. Additionally, the description of Figures 4g-i precedes that of a-f, so the order of the panels should be reversed.
2. The introduction of new metrics for evaluating mitochondrial/crista segmentation is valuable. Therefore, it is essential to clarify how the mitochondrial area is defined for NRMSE_{mito}, PSNR_{mito}, and SSIM_{mito}, as well as how the cristae area is defined for the cristae-specific metrics.
3. Although images with unmeasurable cristae have been excluded from the analysis, the cristae in the raw image in Figure 6a are not visible. To ensure a fair comparison, an image with visible cristae should be included.
4. While the authors conclude that their method outperforms existing techniques, comparisons, especially with existing deep learning methods, are still insufficient to make this claim. It is required to include a comparison with well-established image super-resolution algorithms such as ESRGAN (recently applied to STORM images; <https://www.nature.com/articles/s41467-023-38452-2>) and, if possible, state-of-the-art algorithms like HAT-L (<https://github.com/xpixelgroup/hat>). If such comparisons are not feasible, provide the reasons for their exclusion.
5. Supplementary Figure 4 indicates that the proposed loss and sampling methods do not significantly improve most metrics, except for a slight increase in PSNR. Notably, there is no substantial difference compared to SCoP1,1. If SCoP1,1 is equivalent to DSSIM loss, the contribution of the new loss appears minor, and it has no impact on SSIM. For NRMSE, DSSIM loss performs better. Authors should discuss the contribution of the new loss in the improvement by DeepCristae. Also, if SCoP1,1 is a new metric, the authors should discuss it in the main text and compare it with DSSIM loss.
6. Based on the descriptions and figures, it appears that some portions within the patches used for training may overlap with those used for validation, raising the possibility of data leakage. To ensure a fair evaluation of the learning process, the patch sampling method should be refined to avoid overlap between images in the training and validation sets.
7. While the authors state that 'no meaningful "hallucination" is observed' in line 240, the evaluation of artifacts generated by DeepCristae is not entirely clear. Given the differences between the DeepCristae images and ground truth in images and plots in Figure 4, the current assessment of the artifacts is not sufficient. Further analyses or a proper explanation for the differences from ground truth should be provided.
8. The variance of the cristae intervals in DeepCristae appears significantly smaller compared to HR STED in Figure 5f. Previous electron microscopy studies have shown significant variance in cristae intervals. Thus, the reduced variance could be due to incorrect cristae restoration. This point should be discussed in more detail.

Minor points

- Fan et al., 2021 (PMID: 33686300) should be cited since it employs a deep learning architecture similar to the one used in this study.
- It is unclear what the GT masks used for SSIM_{cristae} look like; an example image should be provided.
- What do the red circles labeled as 'NS' in Figure 6b and d represent?
- Line 864: 'Deepcristae' should be written as 'DeepCristae.'
- Line 264: '90nm' should be written as '90 nm.'
- The labeling of Figure 4l appears stretched.
- LLSM should be spelled out the first time it is mentioned.
- Abbreviations in the movies should be spelled out."

Reviewer #3

(Remarks to the Author)

Summary:

Papereux et al. have developed a novel convolutional neuronal network (DeepCristae) to reconstitute mitochondrial cristae structure from low spatial resolution microscopy images.

The algorithm aims to restore the cristae information from fast acquisition STED images and other light microscopy techniques (AiryScan, LLSM, Live-SR) which traditionally lack this ultrastructural information. This should enable long-term, fast and live imaging while still being able to retain ultrastructural information of mitochondria.

DeepCristae certainly holds the potential to overcome the temporal limitation of traditional super-resolution techniques and restore high resolution information from different imaging modalities. This would enable also research groups without access to state-of-the-art microscopes to resolve mitochondrial ultra-structures. Nevertheless, there are some major concerns on the methodological approach as well as on to what degree this approach really advances the state-of-the-art to obtain biologically meaningful results.

Major comments:

- 1.) One main concern is that the description of the workflow is for the reader hard to understand and pieces of information are

spread over the main text, materials section and figures. In particular the section explaining the selection of training data and ground truth (Fig. 1) lacks clarity and detailed explanation.

2.) Several points lack references to the corresponding Figure (e.g. line 230ff to Fig. 4 a-f, Fig. 7). Statements often lack clarity and further explanation (for examples, see "Minor comments").

3.) The training of the algorithm should, in my view, ideally not be performed on "degraded" high-resolution images, but on actual low-resolution images (similar to what is shown in Fig. 5). Otherwise, the information of the RAW data will still be contained in the LR images. At least the authors should at least discuss this aspect.

4.) The HR STED images in Fig.4 used as "ground truth" are not of very high quality compared to the state of the art (Lui et al., PNAS 2022; Stephan et al., Sci Rep 2019). The authors should also provide a quantification of cristae density and size from the Deep Cristae reconstructions and from high quality STED images (or correlated EM as in Stephan et al., Sci Rep 2019, Fig. 2).

5.) In the same line, the average cristae distance mentioned in line 266 and Fig. 5 are larger than what has been reported previously. Also when comparing the reconstructed images from DeepCristae (e.g. Fig. 4) with previously published STED and EM images, it seems as the algorithm does not recognize all cristae, but only parts with high signal to noise. Although it is certainly an improvement that some cristae can be resolved, I am not certain if this is a sufficient improvement to make quantitative measurements, where changes in cristae density might only be in the range of 10-30%.

6.) In the last section, the authors use their approach to investigate mitochondrial cristae rearrangements during organelle contacts. However, it is unclear to me what the information gain and extracted findings are from these experiments. Figure 7 lacks quantifications and the panels of the figure are not always referred to in the text. It is also unclear to me where the arrows are pointing to.

Minor comments:

1) In the second part of the abstract (26 ff) lacks clarity. The authors give a preview in details of the algorithm's development. However, due to vague formulation like "meaningful measurement" and quick jumping from one step to another the reader loses track of the main message of the paper.

2) In general, the text should be revised for grammatical mistakes and unclear formulation (e.g., line 91ff, line 142ff, line 142ff, 188f)

3) It is unclear what is meant by "night sky pattern" (line 134).

4) In Figure 6 the label RAW for deconvoluted images is misleading and should be clarified in the Figure legend and not only in the main text (line 309 ff).

5) SFig. 6a needs clarification about where the measurements were taken and what the arrows are pointing to.

Version 1:

Reviewer comments:

Reviewer #1

(Remarks to the Author)

I think that the authors have addressed my previous concerns and the paper is ready for publication.

Reviewer #2

(Remarks to the Author)

The revised manuscript adequately addresses this reviewer's concerns.

Reviewer #3

(Remarks to the Author)

The authors have addressed all points raised and have greatly increased clarity of the manuscript. Therefore, we believe that the current manuscript is suitable for publication

Response To Reviewers

In the following, we describe the changes performed to the manuscript entitled “DeepCristae, a CNN for the restoration of mitochondria cristae in live microscopy images” (COMMSBIO-23-2586-T).

First of all, we would like to thank the reviewers for their careful reading of our work and their positive comments, which have been precious for the improvement of the scientific content and the quality of the manuscript.

Following the remarks of the reviewers, we have changed parts of the manuscript to clarify important aspects of our contributions. These parts have been highlighted in **red color** to ease their locations.

In the next section, we give the reviewers’ comments as a reminder, while giving our answers to them right below in **blue color**.

At the end of this document, we also provide a table listing all changes made between the figures in the former and revised manuscripts.

REVIEWER #1

These authors have developed a deep-learning method to restore mitochondrial cristae from low-resolution microscopy images. Their current method demonstrated several critical advantages over the previous methods. 1) They trained their network by designing a training loss that emphasized similarity on mitochondria but not the background. 2) They also show the superiority of their method over existing methods. 3) Most importantly, they are among the first to address the possible hallucinations induced by the DL method, which I believe shall be encouraged.

We thank the reviewer for his/her positive assessment.

Overall, I support the publication of this paper in Communications Biology, provided that they address a few minor points (mentioned below) in the revision.

We thank the reviewer for his/her comments that prompted us to improve the quality of our paper.

Minor:

1. In Fig.3, the authors have shown representative examples of DL-assisted cristae reconstruction against GT. The image qualities of 3b and 3C were quite impressive, while the 3d and 3e were marginal due to the poor quality of the GT. I wonder whether the authors may design metrics to describe quantitative information regarding the cristae, such as their numbers and sizes. If that could be done, it could be extended to be a more meaningful criterion for evaluating mitochondria.

We agree that the quality of the GT in Figs. 3d and 3e (in the former version of the paper) are of poor quality. There are two reasons for this. First, our HR STED images contain different levels of noise and blur due to mitochondria outside the focal plane. Second, the Richardson-Lucy algorithm, that we apply on the HR STED images during the training step to enhance mitochondria cristae, is known to create some artifacts in noisy areas. In order to improve the quality and the reading of the paper, we made two major changes:

1) We extracted 26 ROIs of our 9 test HR STED images focusing on areas where the mitochondria are in the focal plane. These high-resolution ROIs form our new test set.

2) All metrics and evaluation were performed in comparison to these native HR ROIs directly (without applying the Richardson-Lucy algorithm).

In the revised manuscript, these changes are described in section Methods - "Generation of the 2D STED dataset - D_{synt} " (lines 667 and 681) and all Figures, Tables and Graphs are updated accordingly.

We agree that it is important to propose quantitative measurements regarding cristae (width, interval). In the former version of the paper such measures were already provided on real images and for several microscopy modalities (live STED, LLSM and Live-SR) (see Figs. 6, 7 and 8 in the revised version of the paper). In the revised manuscript, measures on cristae width and number computed over the test set of D_{synt} are added in Fig. 2 and in Supplementary Fig. 8 to improve the comparison to other methods. In Supplementary Fig. 4, we added measures on cristae intervals to further study the reliability of DeepCristae. Each time statistical significance tests are also provided.

2. Suppl Fig. 3 tested the reliability of image restoration by DeepCristae according to the level of noise and blur in the image. This is quite important, so I suggest it be incorporated into the main text, not supplementary.

Thank you for this valuable comment. This figure is now referred as Fig. 4 in the revised manuscript (main text).

3. The authors stated, "For each ROI, a comparison of normalized intensity profiles between the input LR image, DeepCristae restored image and the ground truth image is performed. A consistency between the cristae restored by DeepCristae and the ones present in the ground truths is observed overall, and no meaningful "hallucination" is observed." This indeed touches on a very important point. I would better understand and appreciate the work if the authors could spend more time elaborating on this point.

For any image restoration method, it is important to question its reliability and in particular whether it hallucinates structures. We agree that the sentence "no meaningful "hallucination" is observed" needs more discussion. In section Results - "Reliability of image restoration by DeepCristae." we removed this sentence and instead clearly explained which patterns can be considered as hallucinations (lines 297-301) and better detailed the experiments and the results that led us to the conclusion that DeepCristae is hallucination-free (under good condition of use) (mainly lines 312-323). We also added quantitative measurements to support this point (Supplementary Fig. 4h of the revised manuscript). Moreover, in the same paragraph, we better point out "fake" cristae obtained if DeepCristae is not correctly used. We also added an experiment (Supplementary Fig. 2c, d) to show that, as you would expect, if DeepCristae is applied to data it was not trained for, results might suffer from hallucinations.

Finally, in the paragraph of the Discussion relating to the investigation of the stability and the limits of DeepCristae, we also added references to the corresponding figures for better clarity of the reading (lines 630, 631 and 639). Note that we split the former section Results - "Robustness and stability of DeepCristae" into two sections "Robustness of DeepCristae with respect to noise, blur and mitochondria scale in the low-resolution images" and "Reliability of image restoration by DeepCristae" in order to better identify the experiments related to the investigation of potential hallucinations.

REVIEWER #2

In this paper, the authors have trained a deep learning super-resolution model specifically for mitochondrial cristae by innovating (1) a novel loss function and (2) patch sampling techniques. They have also enhanced existing image-level metrics to effectively evaluate the super-resolution quality of mitochondria and cristae, reporting that their model, named DeepCristae, outperforms existing methods. DeepCristae was applied to data obtained through various imaging techniques including live imaging on cristae dynamics and Endosome/Lysosome-Mitochondria contacts. While the live imaging results presented in the manuscript are intriguing, and their method appears promising for future analyses of ultrastructural dynamics, there are still several important unresolved issues in evaluating the strengths of their approach.

We thank the reviewer for his/her comments that prompted us to improve the quality of our paper.

Major Points

1. The order in which the experiments are described is confusing. In Figure 2, DeepCristae is applied to D_{synt}, yet the manuscript seems to suggest that Figure 3 is the first figure demonstrating the application of DeepCristae to D_{synt}.

Thank you for this valuable comment. To make it clearer we modified the main text of the Section "DeepCristae quantitatively outperforms state-of-the-art algorithms on the synthetic dataset D_{synt}" (lines 164-185) as well as the legend of the Figs. 1, 2 and 3.

Additionally, the description of Figures 4g-i precedes that of a-f, so the order of the panels should be reversed.

Right. If the former Figs. 4g-i and 4a-f are now in two separated figures, we applied this comment to all figures that did not follow the main text order.

2. The introduction of new metrics for evaluating mitochondrial/crista segmentation is valuable. Therefore, it is essential to clarify how the mitochondrial area is defined for NRMSE_{mito}, PSNR_{mito}, and SSIM_{mito}, as well as how the cristae area is defined for the cristae-specific metrics.

We agree. We added more details in Supplementary Note 2.1 on how the mask of the mitochondria and cristae areas were obtained (lines 186, 188 and 193). We also added the Supplementary Fig. 1 to illustrate these masks.

3. Although images with unmeasurable cristae have been excluded from the analysis, the cristae in the raw image in Figure 6a are not visible. To ensure a fair comparison, an image with visible cristae should be included.

The spatial resolution of LLSM and Live-SR is well established at approximately 300 nm and 140 nm, respectively. In both modalities, the raw images do not clearly depict cristae, which appear blurred and noisy in our image database. The average measured widths of cristae in raw LLSM and Live-SR images are 339 nm and 149 nm, respectively, as shown in Fig. 7 of the revised manuscript. After image restoration, the cristae peaks become more defined, with an average width of 94 nm in LLSM and 87 nm in Live-SR. In addition, we changed the thumbnails of Fig. 7a to make the cristae clearer before and after restoration.

In these experiments, we demonstrate that DeepCristae can reveal cristae with better resolution. Unfortunately, no super-resolution microscopy technique can be simultaneously used in these experiments to confirm the actual presence of cristae recovered by DeepCristae. One option could be STED microscopy, which allows for peak resolution around 100 nm, as shown in Fig. 6 of the revised manuscript. However, using this technique would compromise 3D mitochondrial dynamics due to the high laser intensity, especially at high acquisition rates.

4. While the authors conclude that their method outperforms existing techniques, comparisons, especially with existing deep learning methods, are still insufficient to make this claim. It is required to include a comparison with well-established image super-resolution algorithms such as ESRGAN (recently applied

to STORM images; <https://www.nature.com/articles/s41467-023-38452-2>) and, if possible, state-of-the-art algorithms like HAT-L (<https://github.com/xpixelgroup/hat>). If such comparisons are not feasible, provide the reasons for their exclusion.

Thank you for this suggestion. We added a comparison to ESRGAN (see Fig. 2). In Fig. 2 and in Section “DeepCristae quantitatively outperforms state-of-the-art algorithms on the synthetic dataset D_{synt} ”, we already compare DeepCristae to several state-of-the-art deconvolution methods, including the methods suggested by the reviewers. Note that the denoising methods that cannot improve spatial resolution like Noise2Void and ND-SAFIR were discarded in the revised version of the manuscript for consistency and clarity. We think that adding a comparison to the HAT algorithm is not appropriate here and it cannot be easily learned, as this transformer-based method needs a large amount of training data (and significant computing resources), even when pre-trained on ImageNet.

5. Supplementary Figure 4 indicates that the proposed loss and sampling methods do not significantly improve most metrics, except for a slight increase in PSNR. Notably, there is no substantial difference compared to SCoP1,1. If SCoP1,1 is equivalent to DSSIM loss, the contribution of the new loss appears minor, and it has no impact on SSIM. For NRMSE, DSSIM loss performs better. Authors should discuss the contribution of the new loss in the improvement by DeepCristae. Also, if SCoP1,1 is a new metric, the authors should discuss it in the main text and compare it with DSSIM loss.

We agree that the Supplementary Note 2.3 on the ablation study of the supplementary material lacked clarity. First, in Supplementary Note 2.3 (line 239) and in Supplementary Figure 8 (former Supplementary Figure 4) of the revised version of the manuscript, we added a sentence to clarify that SCoP1,1 is equivalent to DSSIM. Second, in Supplementary Figure 8 of the revised manuscript, we added an experiment (Supplementary Figures 8 b-d) to better demonstrate the improvement of our proposed loss and sampling method: the measurement of cristae widths for 155 cristae measurement from the 26 test images after restoration shows that the best results in term of mean, standard deviation and detection are obtained with our model. We modified Supplementary Note 2.3 to reflect these new results.

6. Based on the descriptions and figures, it appears that some portions within the patches used for training may overlap with those used for validation, raising the possibility of data leakage. To ensure a fair evaluation of the learning process, the patch sampling method should be refined to avoid overlap between images in the training and validation sets.

Thank you for this valuable comment. As you noticed, statistically, some parts of the patches used for training may overlap with those used for validation. In our case, this data leakage is rare, but it is indeed preferable to avoid it, especially if it can improve the learning process. We have therefore modified our code to avoid this data leakage as follows: when splitting between validation and training sets, we simply ensure that patches belonging to the same original image are assigned to the same set (validation or training). We added this comment in the main text in Section Methods - Generation of the 2D STED dataset - D_{synt} (line 679).

To avoid any misunderstanding, we would like to make it clear that the data leakage mentioned here only concerns the validation set, and that the only consequence is that our model could have been improved. There is no data leakage concerning the test set. All the results presented in this paper were obtained from test images that have not been seen by the network, and are therefore valid.

7. While the authors state that 'no meaningful "hallucination" is observed' in line 240, the evaluation of artifacts generated by DeepCristae is not entirely clear. Given the differences between the DeepCristae images and ground truth in images and plots in Figure 4, the current assessment of the artifacts is not sufficient. Further analyses or a proper explanation for the differences from ground truth should be provided.

We agree that the evaluation of artifacts generated by DeepCristae is not entirely clear. In section Results - “Reliability of image restoration by DeepCristae.” we better explained which patterns can be considered as hallucinations (lines 297-301) and detailed the experiments and the results that led us to the conclusion that DeepCristae is hallucination-free (under good condition of use) (mainly lines 312-323). We also added quantitative measurements to support this point (Supplementary Fig. 4h of the revised manuscript). In the

same paragraph, we better point out “fake” cristae obtained if DeepCristae is not correctly used. We also added an experiment (Supplementary Fig. 2c, d) to show that, as you would expect, if DeepCristae is applied to data it was not trained for, results might suffer from hallucinations. Finally, in the paragraph of the Discussion relating to the investigation of the stability and the limits of DeepCristae, we also added references to the corresponding figures for better clarity of the reading (lines 630, 631 and 639). Note that we split the former section Results - “Robustness and stability of DeepCristae” into two sections “Robustness of DeepCristae with respect to noise, blur and mitochondria scale in the low-resolution images” and “Reliability of image restoration by DeepCristae” in order to better identify the experiments related to the investigation of potential hallucinations.

Concerning Fig. 4 of the former manuscript (Supplementary Fig. 4 in the revised version), the difference between the line plots is a delay due to the moving of the mitochondria between the successive acquisition (~30s) of the low-resolution image (image to be restored) and the high-resolution image (denoted as “ground truth”). We agree that calling “ground truth” the HR STED image can be confusing. Throughout the manuscript, we removed the misleading term “ground truth”, especially in this paragraph, and modified the figure accordingly to improve the presentation of the experiment.

8. The variance of the cristae intervals in DeepCristae appears significantly smaller compared to HR STED in Figure 5f. Previous electron microscopy studies have shown significant variance in cristae intervals. Thus, the reduced variance could be due to incorrect cristae restoration. This point should be discussed in more detail.

Thank you for this comment. We carefully re-analyzed the cristae intervals estimated on HR-STED. The new results are reported in Figure 6f. These new results confirm that DeepCristae is able to provide the same cristae intervals (with the same variability) from FAST STED images as those obtained with HR (early) STED images.

In fact, when we looked more closely at our measurements, we found some issues with the selection of the early HR STED in former Fig. 5f. This modality of STED, as we have demonstrated, increases mitochondrial size over time and enhances photobleaching, which significantly impacts ridge separation. Therefore, in this new figure, we only selected mitochondria from the first time points for (early) HR-STED, and ensured we chose the healthiest ones available.

Concerning the variance in cristae intervals estimated from electron microscopy images, we note that it is generally very difficult to compare measurements obtained from fluorescence images with those from high-resolution electron microscopy techniques. Additionally, the variance may be caused by differences in cell preparation, the microscopy image acquisition process, and cristae reconstruction (using DeepCristae in our case). Finally, Correlative Light Electron Microscopy (CLEM), which combines two different microscopy modalities, has been specifically developed to address this issue, but it typically requires a complex setup, in this case based on STED and EM modalities. Such an approach will be valuable for confirming these preliminary findings in future work.

Minor points

- Fan et al., 2021 (PMID: 33686300) should be cited since it employs a deep learning architecture similar to the one used in this study.

Thank you. This reference has been added in the bibliography section.

- It is unclear what the GT masks used for SSIM_{cristae} look like; an example image should be provided.

Right. As recommended, we added the Supplementary Figure 1 in the revised manuscript that illustrates masks of mitochondria and cristae used to compute the metrics $NRMSE_{mito}$, $PSNR_{mito}$, $SSIM_{mito}$ and $NRMSE_{cristae}$, $PSNR_{cristae}$, $SSIM_{cristae}$.

- What do the red circles labeled as 'NS' in Figure 6b and d represent?

Right. We confirm that these symbols were not explained. In fact, they referred to non-significant (NS) points. Since they are neither used nor helpful, they have been removed from Figure 7 (former Figure 6) for clarity.

- Line 864: 'Deepcristae' should be written as 'DeepCristae.'

This typo error has been corrected.

- Line 264: '90nm' should be written as '90 nm.'

This typo error has been corrected.

- The labeling of Figure 4I appears stretched.

Thank you. We improved the quality of all tables appearing in figures.

- LLSM should be spelled out the first time it is mentioned.

Lattice Light Sheet Microscopy (LLSM) is now introduced for the first time in the main text in Section "DeepCristae restores 3D+time images of mitochondria cristae by using intermediate high-resolution and diffraction limited microscopy" (line 427) and also in the legend of Figure 1.

- Abbreviations in the movies should be spelled out."

As suggested, we provided the meaning of the abbreviations LLSM, PMDR and PKMO shown in movies in the legends of movies (see Supplementary Material, last page). Additionally, we included the full meaning of LLSM in the movies.

REVIEWER #3

Summary: Papereux et al. have developed a novel convolutional neuronal network (DeepCristae) to reconstitute mitochondrial cristae structure from low spatial resolution microscopy images.

The algorithm aims to restore the cristae information from fast acquisition STED images and other light microscopy techniques (AiryScan, LLSM, Live-SR) which traditionally lack this ultrastructural information. This should enable long-term, fast and live imaging while still being able to retain ultrastructural information of mitochondria.

DeepCristae certainly holds the potential to overcome the temporal limitation of traditional super-resolution techniques and restore high resolution information from different imaging modalities. This would enable also research groups without access to state-of-the-art microscopes to resolve mitochondrial ultra structures.

We thank the reviewer for his/her positive assessment.

Nevertheless, there are some major concerns on the methodological approach as well as on to what degree this approach really advances the state-of-the-art to obtain biologically meaningful results.

Major comments:

1.) One main concern is that the description of the workflow is for the reader hard to understand, and pieces of information are spread over the main text, materials section and figures. In particular, the section explaining the selection of training data and ground truth (Fig. 1) lacks clarity and detailed explanation.

The manuscript has been revised and slightly re-organized. We modified section Method – “Generation of the 2D STED dataset D_{synt} ” and the legend of Fig. 1 to make them self-content. We also added Supplementary Figure 1 to visualize the masks of mitochondria and of cristae which may help to understand section Method – “Image patch sampling for the training step”.

2.) Several points lack references to the corresponding Figure (e.g. line 230ff to Fig. 4 a-f, Fig. 7). Statements often lack clarity and further explanation (for examples, see “Minor comments”).

In the revised manuscript, we added references to Figures through the main text when it is appropriate.

3.) The training of the algorithm should, in my view, ideally not be performed on “degraded” high-resolution images, but on actual low-resolution images (similar to what is shown in Fig. 5). Otherwise, the information of the RAW data will still be contained in the LR images. At least the authors should at least discuss this aspect.

We agree that ideally, it would be preferable to train DeepCristae using real low-resolution images, as long as they can be paired to a ground truth image needed to compute the training loss. However, in our context, the acquisition of a pair of high- and low-resolution images at the exact same time point is not technically possible. This is because mitochondria are living organelles in cells, which move quite quickly, so that displacements and deformations of mitochondria are necessarily observed between two successive 2D acquisitions (see Supplementary Fig. 4a-c of the revised manuscript). This is one of the main difficulties of our application that we overcame by creating the dataset D_{synt} that mimic the contents of real low-resolution images.

The information contained in the raw data (the HR images) is indeed still contained, in a degraded way, in our “synthetic” LR images I_{syntLR} . However, this is not a problem since we want to restore signals that are hidden or degraded in the LR images, not “invent” signals. Actually, with a pair of real high- and low-resolution images, the same remark would also apply: LR signals are mainly noisy and blurred versions of HR signals. In microscopy, it is common practice for conventional and deep learning methods to be trained and/or evaluated on synthetic data, as we did, by appropriately blurring and adding noise to a reference

image. Moreover, the good results obtained by applying our trained DeepCristae on real unseen low-resolution images (Figures 6c and 7 of the revised manuscript) suggest that our training is satisfactory.

4.) The HR STED images in Fig.4 used as “ground truth” are not of very high quality compared to the state of the art (Lui et al., PNAS 2022; Stephan et al., Sci Rep 2019). The authors should also provide a quantification of cristae density and size from the Deep Cristae reconstructions and from high quality STED images (or correlated EM as in Stephan et al., Sci Rep 2019, Fig. 2).

Thank you for this valuable comment. We made important efforts to get high-resolution STED images depicting live mitochondria. In the aforementioned references, the protocols, set-ups (different cell types and culture medium for instance) and objectives are slightly different that could justify the apparently lower image quality with HR STED.

In the manuscript, we provided statistics of cristae widths and intervals in Figure 6e-f of the revised manuscript which are very similar to those reported in the literature. We did not address the question of density as it requires more images and an appropriate definition. We agree that objectively quantify density of cristae and compare to results obtained in electron microscopy is valuable research. In the literature (Liu et al. [<https://www.pnas.org/doi/full/10.1073/pnas.2215799119>], Stephan et al. [<https://www.nature.com/articles/s41598-019-48838-2>]), many different cell types have been used: Cos-7, U2OS, HeLa... We here use RPE1 cells. It is clear that depending on cell type and their metabolic state, variation in cristae density as well as mitochondria shape and size will vary significantly. A study more focused on this aspect should be of interest and the use of DeepCristae might be helpful in this context. Thank you for this valuable comment.

5.) In the same line, the average cristae distance mentioned in line 266 and Fig. 5 are larger than what has been reported previously. Also, when comparing the reconstructed images from DeepCristae (e.g. Fig. 4) with previously published STED and EM images, it seems as the algorithm does not recognize all cristae, but only parts with high signal to noise. Although it is certainly an improvement that some cristae can be resolved, I am not certain if this is a sufficient improvement to make quantitative measurements, where changes in cristae density might only be in the range of 10-30%

Thank you for your valuable comment. The comment regarding the average cristae distance (former Fig. 5f) being larger than previously reported has also been noted by Reviewer 2. We re-analyzed the cristae intervals measured using HR-STED, and the updated results are shown in current Figure 6f, which are now more consistent with the literature. Upon closer examination, we found issues with selecting early HR-STED images in Fig. 6f (formerly Fig. 5f). HR STED modality can increase mitochondrial size over time and enhance photobleaching, impacting ridge separation. Thus, in the revised figure, we focused on the first time points for HR-STED and selected only the healthiest mitochondria.

We agree that expanding our training data with previously published STED and EM images is an interesting approach, though it requires standardization in sample preparation, microscopy parameters and cell types. Additionally, we used 2D live imaging for STED, which presents the challenge of maintaining focus due to the highly dynamic nature of mitochondria. An interesting approach could be to use 3D STED which unfortunately was not available to us. Overall, we believe our manuscript already provides a proof-of-concept for improvements that can be adapted in future studies using other modalities, including correlative electron microscopy applied to mitochondrial cristae.

6.) In the last section, the authors use their approach to investigate mitochondrial cristae rearrangements during organelle contacts. However, it is unclear to me what the information gain and extracted findings are from these experiments. Figure 7 lacks quantifications and the panels of the figure are not always referred to in the text. It is also unclear to me where the arrows are pointing to.

In Figure 8 (former Figure 7), our intention is to show preliminary results and the potential of DeepCristae to investigate the spatial re-arrangements of cristae during the dynamic interactions between mitochondria and endo/lysosomes in 3D live cell imaging. In this revised manuscript, we included a supplementary figure (Supplementary Figure 6 and described in the main text lines 544-548) to study the contact between lysosomes and mitochondria. To our knowledge, observing such dynamics in high-resolution microscopy is

not frequently reported in the literature. Our objective was to demonstrate that our AI-based proof-of-concept can serve as a new method to investigate previous datasets.

We have rearranged the former Figure 7 into the new Figure 8 and Supplementary Figure 5d, e. We also added the quantification of the cristae intervals during endo/lysosome contacts in the new Figure 8. Additionally, we improved the legends and overall presentation. Furthermore, we removed the arrows to avoid confusion.

Minor comments:

1) In the second part of the abstract (26 ff) lacks clarity. The authors give a preview in details of the algorithm's development. However, due too vague formulation like "meaningful measurement" and quick jumping from one step to another the reader loses track of the main message of the paper.

We agree. We improved this part.

2) In general, the text should be revised for grammatical mistakes and unclear formulation (e.g., line 91ff, line 142ff, line 142ff, 188f)

The revised manuscript was corrected in the aforementioned lines and more globally proofread to improve its quality.

3) It is unclear what is meant by "night sky pattern" (line 134).

These artifacts are typically induced by iteratively deconvolving a highly noisy image, especially with the Richardson-Lucy algorithm, which creates spatial patterns in the background. However, in the revised manuscript, the "night sky" artifact is no longer mentioned, as we now quantitatively compare the DeepCristae results with the raw high-resolution STED images.

4) In Figure 6 the label RAW for deconvoluted images is misleading and should be clarified in the Figure legend and not only in the main text (line 309 ff).

Thank you for this remark. The figure legend has been updated accordingly for Fig. 7c, d (formerly Fig. 6c, d) to indicate the use of realignment (deskew) and Richardson-Lucy deconvolution for LLSM data.

5) Supplementary Fig. 6a needs clarification about where the measurements were taken and what the arrows are pointing to.

Thank you for the comment. We decided to remove the arrows from Fig. 8b (formerly Supplementary Fig. 6a) to avoid confusion, and we added a profile line to indicate where the intensity profile was taken.

Description of the changes between the figures in the former and revised manuscripts

Revised manuscript	Former manuscript	Changes	
Figure 1	Figure 1	Legend update: I_syntGT -> I_HR as we now consider as ground truths the native HR STED images.	
Figure 2	Figure 2	 • a & b: update of the illustrations, legends and metrics in relation of our new test dataset and that now the ground truths are the native HR STED images; addition of ESRGAN, a state-of-the-art method, and removal of the denoising methods that cannot improve spatial resolution like Noise2Vois and ND-SAFIR; • c: addition of Fourier Ring Correlations; • d-f: addition of biological measurements (cristae width and number) as well as statistical tests. 	
Figure 3	Figure 3	Update of the illustrations, legends and graphs in relation of our new test dataset and that now the ground truths are the native HR STED images.	
Figure 4	Supplementary Figure 3	Update of the illustrations and graphs in relation of our new test dataset and that now the ground truths are the native HR STED images.	
Figure 5	Figure 4 g-l	Update of the metrics in relation of our new test dataset and that now the ground truths are the native HR STED images.	
Figure 6	Figure 5	 • a: update of the illustrations; • f & g: update of the values of the cristae intervals for Early HR STED. 	
Figure 7	Figure 6	 • a: update of the illustrations to make the cristae clearer before and after restoration; • b & d: removal of the NS circles and legend update; • e: contrast improvement. 	
	a	Figure 7 (a-b)	Keeping two ROIs out of four and fewer time points for a better visualization.

Figure 8	b	Supplementary Figure 6a	None
	c	Supplementary Figure 6b	None
	d	Supplementary Figure 6c	None
Supplementary Figure 1		New figure	
Supplementary Figure 2	a-b	Supplementary Figure 2 (a-b)	Update of the illustrations and graphs in relation of our new test dataset and that now the ground truths are the native HR STED images.
	c-d	New Figure	
Supplementary Figure 3		Supplementary Figure 2 (c-h)	Update of the metrics in relation of our new test dataset and that now the ground truths are the native HR STED images.
Supplementary Figure 4		Figure 4 (a-f)	 • a-c: Legend update: Input -> LR STED & GT -> HR STED & addition of time frame (in seconds); Update of the illustrations. • d-g: update of the illustrations, legends and graphs in relation of our new test dataset and that now the ground truths are the native HR STED images • h: addition of biological measurements (cristae intervals) as well as statistical tests.
Supplementary Figure 5	a-c	Supplementary Figure 5	None
	d-e	Figure 7 (c-d)	Keeping two ROIs out of three and fewer time points for a better visualization.
Supplementary Figure 6		New figure	
Supplementary Figure 7		Supplementary Figure 1	Legend update: I_syntGT -> I_HR as we now consider as ground truths the native HR STED images.
Supplementary Figure 8		Supplementary Figure 4	 • Removal of the illustrations; • a: update of the metrics in relation of our new test dataset and that now the ground truths are the native HR STED images; Clarification that SCoP_1,1 = DSSIM.

		• b-d: addition of biological measurements (cristae width and number) as well as statistical tests.
--	--	---

Response To Reviewers

In the following, we respond, in blue color, to the comments of the reviewers of the manuscript entitled "DeepCristae, a CNN for the restoration of mitochondria cristae in live microscopy images" (COMMSBIO-23-2586A).

The comments of the reviewers did not result in any changes to the manuscript.

We would like to thank the reviewers for their careful reading of our work and their positive comments.

REVIEWER #1

I think that the authors have addressed my previous concerns and the paper is ready for publication.

We thank the reviewer for his/her positive assessment.

REVIEWER #2

The revised manuscript adequately addresses this reviewer's concerns.

We thank the reviewer for his/her positive assessment.

REVIEWER #3

The authors have addressed all points raised and have greatly increased clarity of the manuscript. Therefore, we believe that the current manuscript is suitable for publication.

We thank the reviewer for his/her positive assessment.